# GameArena: Evaluating LLM Reasoning through Live Computer Games

**Lanxiang Hu**[*][1], **Qiyu Li**[*][1], **Anze Xie**[*][1], **Nan Jiang**[1], **Ion Stoica**[2], **Haojian Jin**[†][1], **Hao Zhang**[†][1]
[1]University of California, San Diego
[2]University of California, Berkeley

## Abstract

Evaluating the reasoning abilities of large language models (LLMs) is challenging. Existing benchmarks often depend on static datasets, which are vulnerable to data contamination and may get saturated over time, or on binary live human feedback that conflates reasoning with other abilities. As the most prominent dynamic benchmark, Chatbot Arena evaluates open-ended questions in real-world settings, but lacks the granularity in assessing specific reasoning capabilities. We introduce GameArena, a dynamic benchmark designed to evaluate LLM reasoning capabilities through interactive gameplay with humans. GameArena consists of three games designed to test specific reasoning capabilities (e.g., deductive and inductive reasoning), while keeping participants entertained and engaged. We analyze the gaming data retrospectively to uncover the underlying reasoning processes of LLMs and measure their fine-grained reasoning capabilities. We collect over 2000 game sessions and provide detailed assessments of various reasoning capabilities for five state-of-the-art LLMs. Our user study with 100 participants suggests that GameArena improves user engagement compared to Chatbot Arena. For the first time, GameArena enables the collection of step-by-step LLM reasoning data in the wild. Our code is available at https://github.com/lmgame-org.

## 1 Introduction

Recent large language models (LLMs) (e.g., OpenAI's o1 model) have made great strides in performing complex reasoning (OpenAI, 2024a; Luo et al., 2024; Ye et al., 2024a;b; Lightman et al., 2023), but evaluating LLM reasoning capabilities remains challenging. Current methods typically use curated test suites of coding and math problems to assess different reasoning skills (Hao et al., 2024; bench authors, 2023; Cobbe et al., 2021). However, these static benchmarks are vulnerable to data contamination issues (Sainz et al., 2023; Rajore et al., 2024) and can quickly become saturated (Kiela et al., 2021; Perlitz et al., 2024), limiting their effectiveness.

Researchers have been exploring alternative dynamic evaluation approaches. The most prominent attempt is Chatbot Arena (Chiang et al., 2024), a crowdsourced platform that asks users to input arbitrary prompts and compare the responses generated by two different models, producing overall ranking scores based on pairwise comparisons. Nevertheless, these ranking scores may not accurately reflect the models' reasoning capabilities, as they depend on human preferences that can be confounded by various factors. For example, Li et al. (2024) found that response style has a strong effect on the leaderboard performance since humans usually favor detailed and "beautiful" responses, which do not relate to reasoning abilities. Additionally, Chatbot Arena could not assess specific reasoning capabilities such as inductive and deductive reasoning, due to its reliance on open-ended user input and lack of control over the reasoning skills required for the prompts.

We introduce GameArena, a novel dynamic benchmark suite designed to evaluate LLM reasoning through three interactive games with humans. GameArena leverages live human-AI interactions to prevent benchmark saturation. In contrast to Chatbot Arena, GameArena confines the human-AI interactions using designated gaming rules, allowing for better control of the interactions toward

---

[*]Equal contributions. Part of work was done during Lanxiang's internship at Snowflake.
[†]Correspondence to haozhang@ucsd.edu, h7jin@ucsd.edu.

testing LLM's fine-grained reasoning capabilities, including deductive, inductive, abductive, and multi-hop reasoning (Huang & Chang, 2023; Seel, 2011). Additionally, embedding evaluation tasks into games enables a more engaging and enjoyable experience for users, encouraging sustained participation in the evaluation process.

Designing effective games for evaluating LLM reasoning is challenging because they need to provide high-quality reasoning data while keeping gameplay engaging for human players. Therefore, instead of creating entirely new games, we integrate LLMs into existing games, i.e., Akinator, Taboo, and Bluffing, that are fun and require complex reasoning through multi-turn conversations. In these games, the LLM attempts to deduce or extract specific information by synthesizing human responses over multiple rounds, which requires nuanced reasoning capabilities beyond simple natural language processing. For example, in the Akinator game, the LLM asks a series of binary questions to guess the object the human player is thinking of. To win the game, the LLM must demonstrate strong deductive reasoning by analyzing players' responses and narrowing down possibilities to derive the correct object.

In addition to the game outcome, the interaction process also provides rich evaluative potential. During gameplay, the LLM develops a step-by-step reasoning trajectory intrinsically, with each round's decision or action serving as a data point that can be analyzed to assess its reasoning capabilities. To uncover the LLM's hidden reasoning process, we retrospectively analyze the gaming data by asking the LLM to reveal its intermediary thought process at each step.

We evaluated five state-of-the-art LLMs with GameArena. To reduce potential prompt bias, we developed five optimized system prompts using DSPy (Khattab et al., 2024) and randomly selected one for each game session. We collected more than 2000 game sessions and analyzed the data to score each LLM based on capability-specific evaluation metrics. We found Claude 3.5 Sonnet outperformed GPT-4o in most evaluation metrics for reasoning. Mistral-large fell short in most reasoning capabilities and often failed to follow game rules and make predictions on game secrets in all three games.

We conducted comparative studies between GameArena and Chatbot Arena. GameArena was found to be more effective in data collection, with over 85% of game sessions from GameArena containing useful data for evaluating LLM reasoning. In contrast, only less than 5% of conversation sessions from Chatbot Arena yielded meaningful votes for LLM evaluation. Our user study with 100 participants suggested that users preferred evaluating LLMs through games, were more satisfied with the experience, and showed greater interest in participation in GameArena compared to Chatbot Arena.

In this paper, we make the following contributions:

- We build GameArena, the first dynamic benchmark for evaluating specific LLM reasoning capabilities using games that collect human-labeled step-by-step reasoning data.
- We introduce a new set of data analysis techniques to evaluate LLM reasoning capabilities while controlling variables like system prompt and style variances.
- We demonstrate the effectiveness of GameArena in comparison with existing baselines when evaluating different capabilities.
- We will release our gaming data for future research.

## 2 GAMES IN GAMEARENA

GameArena aims to evaluate LLM reasoning capabilities through live games with humans while keeping participants entertained and engaged. GameArena consists of three games, Akinator, Taboo, and Bluffing. For each game, we instruct the LLM to interact with humans according to the game rules. Since these games are designed to require strong reasoning skills, the game performance of LLMs reflects their reasoning capabilities. We first explain the games and how they are associated with LLM reasoning capabilities, followed by a formal formulation of the game workflow.

### 2.1 GAME DESIGN

**AI Akinator game** (Fig. 1) follows the same rule of the online Akinator game (Akinator, 2024). An LLM attempts to determine what object the player is thinking of by asking up to 20 yes or no

questions. The goal of the LLM is to guess the answer correctly with the fewest number of questions possible. We provide a detailed description of game rules in Appendix A.1.1.

This game presents a controlled setting with a limited number of guesses to evaluate the LLM's deductive and multi-hop reasoning skills. **Deductive reasoning** (Creswell et al., 2022; Han et al., 2022; Saparov & He, 2023) involves drawing a specific conclusion from a chain of prior premises. In this game, each factual question-answer pair, from general descriptions to specific details, serves as a premise to narrow down possible options. All the premises logically lead to the most probable secret object the user has in mind. **Multi-hop reasoning** (Yang et al., 2018; Ho et al., 2020) entails connecting multiple pieces of information or reasoning steps to solve a problem. In the Akinator game, the LLMs must employ multi-hop reasoning to synthesize and connect information from multiple rounds. A strong LLM uses multi-hop reasoning to ask insightful questions based on prior knowledge and employs deductive reasoning to quickly narrow down possibilities for each answer.

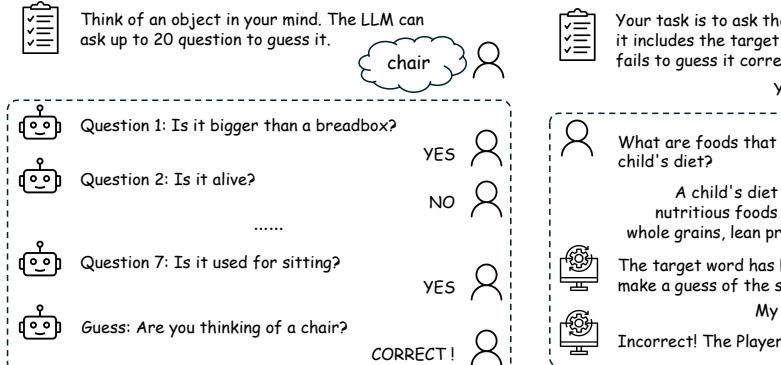

Figure 1: In the Akinator game, a player thinks of an object, and the LLM asks a series of yes-or-no questions to guess the object.

Figure 2: In the Taboo game, a player has a taboo word that they need to prompt the LLM to say without revealing what the word is.

**AI Taboo game** (Fig. 2). The goal of the human player is to prompt the LLM with questions that will lead the model to say the target word, even though the LLM has been instructed to guess the word and avoid uttering it in its response. We describe the game rules in Appendix A.1.2.

This game evaluates LLM's abductive and multi-hop reasoning skills. **Abductive reasoning** (Jung et al., 2022; Seel, 2011) involves generating a possible explanation from incomplete information. Throughout the game, the player discloses fragmented and ambiguous information related to the secret word, and the LLM needs to reason the most possible target word based on the limited clues. In rare cases, the LLM's response may directly include the target word, allowing it to guess correctly in the next round. However, most cases require complex reasoning. Strong LLMs use multi-hop reasoning to extract clues from previous prompts and apply abductive reasoning to infer the most likely target word. A detailed example is provided in Appendix E.3.

**AI Bluffing game** (Fig. 3). The human player aims to convince an LLM model that they are making a true statement. The LLM can ask a player at most five questions to determine whether the player is lying. The rule of the Bluffing game is outlined in Appendix A.1.3.

This game challenges LLMs to detect deception by proposing strategic questions and identifying subtle flaws in the responses of human players using inductive and multi-hop reasoning. **Inductive reasoning** (Misra et al., 2022; Yang et al., 2022) involves making predictions or drawing conclusions based on a set of observations, where observations are typically empirical and not guaranteed to be true. In this game, the LLM observes multiple question-answer pairs to make a final prediction about the truthfulness of the user's statement. Ideally, a good LLM utilizes multi-hop reasoning to connect clues from existing information, develop effective questioning strategies that maximize information gain and apply inductive reasoning to draw a well-informed conclusion.

## 2.2 GAME FORMULATION

In GameArena, each game session starts with randomly pairing a game $\mathcal{G}$ with a state-of-the-art LLM $f_k$ parameterized by $\Theta_k$ from a list of candidate models $\{f_k\}$ and one of the candi-

Make a statement and try to convince an LLM that your statement is true.

Statement: I have been to space.

Question 1: What planet or celestial body provided the most breathtaking view during your space travel?

I just saw the earth and came back.

Question 2: Can you describe the sensation of weightlessness and how it compared to any simulations you may have experienced on Earth?

It was crazy - everything was floating in the air, and I felt extremely light.

......

Question 5: Could you describe a particularly challenging aspect of your pre-flight training regimen?

I had to sit in a spinny thing with really strong g force.

I believe your statement is: False because ......

CORRECT !

Figure 3: In the Bluffing game, the player makes a statement and responds to a series of questions from the LLM as if the statement is true, trying to trick the LLM into believing it.

date system prompts be $\mathbf{x}_{k,p}$. Let the maximum round limit for $\mathcal{G}$ be $N$, the ground truth be $\mathbf{g}$. For each turn $i$, let $\mathbf{x}_i$ be user's input, $f_k$'s generation at each time step $t$ can be denoted as $\mathbf{y}_{i,t} = f_k\left([\mathbf{x}_{k,p}, \mathbf{x}_1, \mathbf{y}_1, \cdots, \mathbf{x}_i, \mathbf{y}_{i,[1:t-1]}]; \Theta_k\right)$. We denote the complete output of length $T$ is $\mathbf{y}_i = \mathbf{y}_{i,[1:T]}$. For each $\mathbf{y}_i$, there are two possible output types. The first one is an ordinary question or answer in response to user's input to advance the gaming process. The other is a game secret prediction which requires the model to either guess the target word in Akinator and Taboo, or make a prediction on the user statement's truthfulness. We denote the first type as $\mathbf{y}_i$ and the second type as $\mathbf{p}_i$. The game ends whenever the maximum round limit is reached or the user verifies $\mathbf{p}_i = \mathbf{g}$. A winner can therefore be decided.

## 3 Evaluating LLM Reasoning through Games

### 3.1 Retrospective Analysis

In order to reveal the model's intermediate thought process throughout the game session and assess various reasoning capabilities demonstrated by the model's hidden chain-of-thought (Wei et al., 2022), we conduct a retrospective analysis based on the chat history we collect from the game sessions. To reenact the game trajectory and capture the model's hidden thoughts during the game sessions, we first keep the system prompt, game history, and inference parameters identical to those used during the original gameplay. Then, we prompt the model to generate intermediate outputs, and conduct quantitative and qualitative analysis based on those outputs.

**Akinator.** The goal of Akinator's retrospective analysis is to assess the model's multi-hop reasoning and deductive reasoning capabilities (Yang et al., 2018; Creswell et al., 2022). After each question-answer round, the model is asked to generate a ranked list of possible objects from its hidden thoughts during the gameplay. The object list prioritizes those it considers most likely based on its reasoning process. For each game session, we then inspect the ranked object lists from all rounds to determine whether the secret object has been correctly identified and its position in the rankings.

**Taboo**. The retrospective analysis of Taboo aims to evaluate the model's multi-hop reasoning and abductive reasoning capabilities (Yang et al., 2018; Jung et al., 2022). Similar to Akinator, the model is prompted to generate a list of possible target words based on all previous game history after each user prompt. The list of possible words is ranked from the most to least likely by the model.

**Bluffing**. The Bluffing game's retrospective analysis aims to reflect the model's multi-hop reasoning and inductive reasoning capabilities (Yang et al., 2018; Misra et al., 2022). After the user's statement and each question-answer round, the model is queried to predict the truthfulness of the statement based on the existing game history. For each prediction, the model selects from five possible choices: 1) True, 2) Possibly true, 3) Unknown, 4) Possibly false, or 5) False, and the model always starts from a neutral position with "Unkown" as its default choice. The intermediate predictions reveal the

model's reasoning process and allows us to assess not only its ability to draw correct conclusions but also its capacity to ask strategic questions that consistently advance the game.

## 3.2 EVALUATION METRICS

### 3.2.1 OUTCOME METRICS

An intuitive set of metrics from our gaming data is the gaming outcome. In Akinator and Bluffing, users provide feedback on whether the model made a correct guess, which determines the winner. In Taboo, a ruled-based keyword detection mechanism is used in Taboo to check whether the user has violated the game rules or the model has lost by saying the word. For each game, we track the average number of rounds and the winning rate by calculating $\sum_i r_i / N_{\mathcal{G}}$ and $\sum_i I_i / N_{\mathcal{G}}$, where $N_{\mathcal{G}}$ is the total number of game sessions for game $\mathcal{G}$, $r_i$ is the number of gaming rounds for each game session, and $I_i \in \{0, 1\}$ is a binary indicator function of whether the LLM won that session.

### 3.2.2 PROCEDURAL METRICS

In retrospective analysis, for any turn $i$, as long as $\mathbf{y}_i \neq \mathbf{g}$, we obtain intermediate results by fixing all the game history $[\mathbf{x}_{k,p}, \mathbf{x}_1, \mathbf{y}_1, \cdots, \mathbf{x}_i, \mathbf{y}_i]$, and ask the model to justify its response $\mathbf{y}_i$ or $\mathbf{p}_i$ using quantifiable data, such as an object list or a truthfulness rank. Consider the justification as $\mathbf{y}_i'$, we iterate over each game session to collect step-by-step reasoning data $[\mathbf{y}_1', \cdots, \mathbf{y}_N']$, from where *procedural metrics* will be calculated for quantitative evaluations and rankings in Table 3.

We developed several metrics to evaluate each reasoning capability (Table 1) based on the model's intermediate outputs from retrospective analysis. Below, we describe how these metrics are mapped to each reasoning capability and how they are computed.

**Akinator**. *Recall rate*, *top-k recall rate*, and *disparity ratio* reflect the model's multi-hop reasoning capability in Akinator. The recall rate metrics determine how well the model utilizes preceding information by measuring how often the correct object appears in object lists retrieved for each game session during retrospective analysis. In each game session, *recall rate* represents the proportion of model-proposed object lists that contain the correct object. *Top-5 recall rate* and *top-10 recall rate* represent the proportion of object lists where the rank of the correct object is within the top 5 and 10 positions. Higher recall rates imply higher utilization of preceding information, indicating better multi-hop reasoning skills.

Multi-hop reasoning is also reflected through the question quality in Akinator. We use *disparity ratio* to assess the information gain of a question. In Akinator, high-quality questions should divide the possibilities evenly into two halves (Sasson & Kenett, 2023). Thus, we compute the disparity ratio based on how evenly a model-generated yes/no question can divide the object list proposed one round prior into two categories. Evenly divided object lists have a lower disparity ratio, suggesting higher information gain.

$$\text{disparity ratio} = \frac{|\text{size}_{\text{yes}} - \text{size}_{\text{no}}|}{\text{size}_{\text{object\_list}}} \tag{1}$$

We use *average first appear round* and *average final rank* as indicators for model's deductive capability in Akinator game sessions. *Average first appear round* refers to the average round number in which the model adds the correct object into the object list. *Average final rank* is the rank of the correct object in the object list proposed after the last round of conversation. Lower *average first appear round* means the model is able to draw conclusion on the correct object with less premise. Lower *average final rank* shows more confidence on the model's choice of correct object. These two metrics both reflect the model's efficiency on deductive reasoning.

**Taboo**. In the Taboo game, we use recall rates to assess the model's multi-hop reasoning skill. The definition and implication of the recall rates here are the same as those used in the Akinator game. We apply the same formulas on the word lists retrieved from retrospective analysis to compute the recall rates for Taboo game sessions.

Abductive reasoning skill is reflected by the metric of *average first appear round* and *average final rank* in Taboo game. *Average first appear round* is the average round number in which the model adds the correct word into the word list. A lower *average first appear round* means the model can efficiently find the correct word based on less interaction with users. *Average final rank* is the rank

of the correct word in the word list proposed after the last round of conversation. A low *average final rank* implies that the model is more confident with its choice of the correct word. Both metrics indicate the model's abductive reasoning skill.

Table 1: The procedural analysis metrics for the Akinator, Taboo, and Bluffing games and their corresponding model capabilities.

| | | |
|---|---|---|
| Akinator | **Deductive reasoning**
first appear round avg, final rank | **Multi-hop reasoning**
recall rate, top-k recall rate, disparity ratio |
| Taboo | **Abductive reasoning**
first appear round avg, final rank | **Multi-hop reasoning**
recall rate, top-k recall rate |
| Bluffing | **Inductive reasoning**
first appear round avg, final rank | **Multi-hop reasoning**
consistency rate, recall rate, top-k recall rate |

**Bluffing**. In the Bluffing game, we use *recall rate* and *consistency rate* to evaluate the model's multi-hop reasoning capability. *recall rate* measures how often the model correctly predicts the truthfulness of the user's statement. *consistency rate* measures whether consecutive predictions consistently move the game forward in the same logical direction, demonstrating the model's ability to build upon previous information and ask strategic questions. We use two metrics, the Spearman's coefficient (Sedgwick, 2014) and the hopping penalty score to measure consistency rate.

Spearman's rank correlation measures the correlation between the round numbers and the "distance" of judgement from the ground truth. At each round, we rank the five retrospective predictions $\mathbf{y}'_i$ as $j_i \in [1, 5]$, corresponding to the 5-level spectrum spanning from "True" to "False". For each game session, Let $D_i$ be the absolute distance between the intermediate prediction and since the ground truth's rank is $g \in [1, 5]$, we can write the expression in Eq. 2. A negative Spearman's correlation suggests that as rounds progress, the distance between the assistant's judgments and the ground truth decreases and the model's prediction gets consistently more accurate.

$$\rho = 1 - \frac{6}{N(N^2 - 1)} \sum_{i=1}^{N} d_i^2, \quad d_i = i - D_i, \quad D_i = |j_i - g| \tag{2}$$

The hopping penalty measures how inconsistent the model's predictions are across consecutive rounds. The smaller the metric is, the better the model is at making consistent progress toward its final prediction, demonstrating stability in questioning strategy and unpredictability in its results. The formula is shown in Eq. 3.

$$\text{hopping penalty} = \frac{1}{N-1} \sum_{i=1}^{N-1} |j_{i+1} - j_i| \tag{3}$$

We use *average first appear round* and *average final rank* as indicators for model's inductive reasoning capability as they reflect how efficiently the model can make correct predictions in the fewest number of rounds. *Average final rank* is averaged across all game sessions, and the rank follows the same definition of $D_i$ in Eq. 2, but only for the final round.

## 4 EXPERIMENTS

### 4.1 SETUP

GameArena evaluates LLM reasoning using gaming data to compare five SOTA models: GPT-4o (OpenAI, 2024b), Claude 3.5 Sonnet (Anthropic, 2024), Gemini-1.5 Pro (Reid et al., 2024), Mistral Large 2 (Mistral, 2024), and LLaMA-3.1 405B-Instruct (MetaAI, 2024). Over a 10-week period from July 2024 to September 2024, we collect a total of 2240 game sessions using Cloud Research (Hartman et al., 2023) for evaluation. We then conduct retrospective data analysis introduced in Section 3 on the gaming data to obtain outcome metrics and procedure metrics for each model.

**Controlling system prompt variance.** In GameArena, we develop a system prompt search and optimization pipeline using DSPy (Khattab et al., 2024). This pipeline uses the chain-of-thought module from DSPy and one of the five models acting as the evaluator. The evaluator guide the model to follow the game rules and judges on response quality. The DSPy optimizer learns to bootstrap and

identify effective system prompts. Each optimizer is optimized from a subset of all our gaming data consisting of 200 game sessions. Given a game and one of the five evaluators, the pipeline searches for an optimal system prompt that maximizes the evaluator's judgment on the model's generated questions or answers at each round, resulting in five highly optimized system prompts per game.

## 4.2 DATA EFFICIENCY

We compared the data efficiency of GameArena and Chatbot Arena by analyzing the useful data rates. We obtained real conversation data from the Chatbot Arena Team for the week of August 26th to September 1st and compared it with the data collected by GameArena during the same period. We considered data as useful if it provides meaningful signals that can be used to evaluate models. In Chatbot Arena, data was deemed useful if it included at least one vote (e.g., for left, right, tie, or both bad). For GameArena, we considered the game session useful if the user completed the entire game round and provided feedback on the game outcome (i.e., win or lose).

We found that 86.9% of the data from GameArena were valid and useful, while only 4% of total conversations in Chatbot Arena provided meaningful votes due to its reliance on voluntary participation. Our results suggest that GameArena could improve user engagement through gamification approaches and collect meaningful data for evaluating LLMs more efficiently.

## 4.3 USER STUDY

**Participant ratings.** We conducted a user study to compare the user experience and willingness to participate in GameArena and Chatbot Arena (Chiang et al., 2024). We recruited 100 participants through Cloud Research (Hartman et al., 2023) and asked them to try both platforms and fill out a survey to provide their feedback. Each participant was randomly assigned a game in GameArena and asked to play at least three rounds. For Chatbot Arena, participants voted on the better responses from two models to their questions. We provided sample questions from MT-Bench (Zheng et al., 2023) for reference, and participants were also free to create their own questions. Participants rated each platform on how much they enjoyed it, how satisfied they were with the experience, and how often they would like to participate on a 5-point Likert scale.

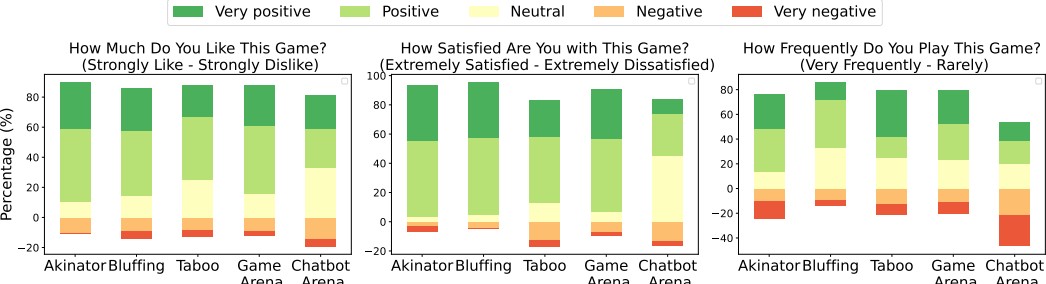

Figure 4: Participant rating distributions across different games in GameArena and Chatbot Arena. A lower bar represents a lower proportion of participants expressing a positive attitude. GameArena showed higher user enjoyment, satisfaction and willingness to participate than Chatbot Arena.

Our user study results suggest that GameArena is more engaging and enjoyable for users (Fig. 4). On average, over 70% of users liked the games in GameArena, compared to only 45% who enjoyed voting in Chatbot Arena. Additionally, over 80% of participants reported satisfaction with gameplay experience in GameArena, compared to less than 40% of users felt satisfied about the experience with Chatbot Arena. Participants also expressed stronger willingness to participate in GameArena, with over half of the participants would like to play these game frequently, whereas interest in Chatbot Arena was more evenly distributed across different participation levels.

## 4.4 RANKING RESULTS

**Model performance on outcome metrics.** The final outcome metrics of the three games are shown in Table 2. The win rates of both claude-3-5-sonnet-20240620 and gpt-4o-2024-08-06 are higher in all three games compared to other models evaluated. The high win rate here reflects the strong overall reasoning ability of the two models.

Table 2: Outcome ranking for each game. Despite with system prompt search and optimization, in many Bluffing games, Mistral-large (Mistral, 2024) fails to follow instructions and make a final prediction on the truthfulness of the users' statements within the 5-round limit. The numerical values are averaged across the five system prompts, with the standard deviation included as the error bar.

| Model | Akinator | | Taboo | | Bluffing | |
|---|---|---|---|---|---|---|
| | Avg. Win Rate | Avg. # Round | Avg. Win Rate | Avg. # Round | Avg. Win Rate | Avg. # Round |
| claude-3-5-sonnet-20240620 | $0.55_{\pm 0.11}$ | $16.61_{\pm 1.75}$ | $0.61_{\pm 0.18}$ | $3.36_{\pm 0.88}$ | $0.67_{\pm 0.13}$ | $6.00_{\pm 0.00}$ |
| gpt-4o-2024-08-06 | $0.49_{\pm 0.13}$ | $16.36_{\pm 0.86}$ | $0.67_{\pm 0.11}$ | $3.19_{\pm 0.34}$ | $0.58_{\pm 0.13}$ | $5.92_{\pm 0.18}$ |
| gemini-1.5-pro | $0.51_{\pm 0.17}$ | $16.57_{\pm 1.49}$ | $0.61_{\pm 0.04}$ | $3.74_{\pm 0.45}$ | $0.60_{\pm 0.18}$ | $5.96_{\pm 0.10}$ |
| llama-3.1-405b | $0.44_{\pm 0.04}$ | $17.15_{\pm 0.66}$ | $0.62_{\pm 0.18}$ | $3.08_{\pm 0.18}$ | $0.44_{\pm 0.22}$ | $5.90_{\pm 0.27}$ |
| mistral-large-latest | $0.02_{\pm 0.04}$ | $19.99_{\pm 0.02}$ | $0.66_{\pm 0.13}$ | $3.43_{\pm 0.57}$ | $0.0_{\pm 0.00}$ | $6.00_{\pm 0.00}$ |

Table 3: The retrospective analysis results of Akinator, Taboo, and Bluffing game sessions hosted by several SOTA LLMs.

| Akinator | Multi-hop reasoning | | | | Deductive reasoning | |
|---|---|---|---|---|---|---|
| Model | Recall Rate (%) | Top 5 Recall Rate (%) | Top 10 Recall Rate (%) | Disparity Ratio (%) ↓ | Avg. First Appear Round | Avg. Final Rank |
| claude-3-5-sonnet-20240620 | **27.83** | **24.58** | **27.26** | 0.57 | **11.63** | **5.31** |
| gpt-4o-2024-08-06 | 20.32 | 17.00 | 19.57 | **0.55** | 11.96 | 6.64 |
| gemini-1.5-pro | 19.79 | 18.56 | 19.79 | 0.58 | 12.12 | 5.44 |
| llama-3-405b | 15.90 | 13.72 | 14.97 | 0.60 | 14.67 | 9.88 |
| mistral-large-latest | 6.46 | 5.90 | 6.46 | 0.62 | 18.07 | 7.11 |

| Taboo | Multi-hop reasoning | | | Abductive reasoning | |
|---|---|---|---|---|---|
| Model | Recall Rate (%) | Top 5 Recall Rate (%) | Top 10 Recall Rate (%) | Avg. First Appear Round | Avg. Final Rank |
| claude-3-5-sonnet-20240620 | 68.85 | **65.82** | 68.85 | **1.56** | **1.48** |
| gpt-4o-2024-08-06 | 55.78 | 53.68 | 55.78 | 2.47 | 3.83 |
| gemini-1.5-pro | 55.57 | 52.58 | 56.31 | 2.55 | 2.44 |
| llama-3-405b | **71.22** | 65.69 | **70.80** | 1.98 | 4.04 |
| mistral-large-latest | 66.30 | 61.34 | 64.88 | 2.05 | 4.89 |

| Bluffing | Multi-hop reasoning | | | Inductive reasoning | |
|---|---|---|---|---|---|
| Model | Recall Rate (%) | Spearman's Coefficient ↓ | Hopping Penalty ↓ | Avg. First Appear Round | Avg. Final Rank |
| claude-3-5-sonnet-20240620 | 14.12 | **-0.39** | **0.20** | 2.46 | **1.47** |
| gemini-1.5-pro | 9.05 | -0.22 | 0.23 | 3.50 | 1.62 |
| gpt-4o-2024-08-06 | 7.64 | -0.15 | 0.44 | 3.00 | 2.06 |
| llama-3-405b | **19.05** | 0.26 | 0.48 | 3.00 | 2.57 |
| mistral-large-latest | 0.00 | -0.20 | 0.27 | N/A | 1.53 |

**Model performance on procedural metrics.** Table 3 presents the procedural metrics designed to reflect the models' reasoning skills from the retrospective analysis of all three games.

**Akinator.** Claude-3-5-sonnet-20240620 performs best on recall rates, *average first appear round*, and *average final rank*, demonstrating stronger multi-hop reasoning and deductive reasoning capabilities compared to other SOTA LLMs being evaluated. Gpt-4o-2024-08-06 also has high performance in the metrics of reasoning capabilities.

**Taboo.** The high performance of claude-3-5-sonnet-20240620 in recall rates, *average first appear round*, and *average final rank* exhibits stronger multi-hop and abductive reasoning capabilities. Notably, llama-3-405b demonstrates strong multi-hop reasoning capabilities with leading recall rates, which differs from the results observed in the Akinator game. This discrepancy may be due to the task differences: in the Taboo game, the model passively responds to user prompts, whereas in the Akinator game, it must actively generate questions to succeed.

**Bluffing.** In many cases, Mistral-large (Mistral, 2024) tends to be very conservative in making predictions. Note that with a large hopping penalty and a high recall rate, LLaMA-3.1 405B tends to make bolder predictions, and its intermediate reasoning outputs tend to be slightly more inconsistent.

## 4.5 RANKING COMPARISONS

**Cross-dataset consistency analysis.** GameArena can be extended to include as many games as possible for evaluating LLM reasoning, following the formulation in Section 2.2. In games like Taboo, we can restrict the set of secret words users are allowed to use, while in others, like Akinator and Taboo, users have the freedom to choose the game secret. It is crucial to quantify the variances in evaluations when the game secret space is large to ensure consistency and fairness across different game setups.

We conducted a controlled experiment to quantify GameArena's evaluation consistency on these games. In this experiment, we asked human experts (graduate students) to hand curate two distinct subsets of gaming data, where the game secrets are sufficiently diverse to both cover a wide range of topics. For each game, the two sets each contains 50 game sessions. We show the outcome metrics for each set in Table 5. We only evaluate the Akinator, and the Bluffing game as Taboo has a restricted game secret space. The results from both subsets show a strong agreement, demonstrating the reliability of our ranking system. More details of the experiment can be found in Appendix B.

Figure 5: Controlled experiment results for Akinator and Bluffing. The numerical values are averaged across the five different system prompts, with the standard deviation included as the error bar.

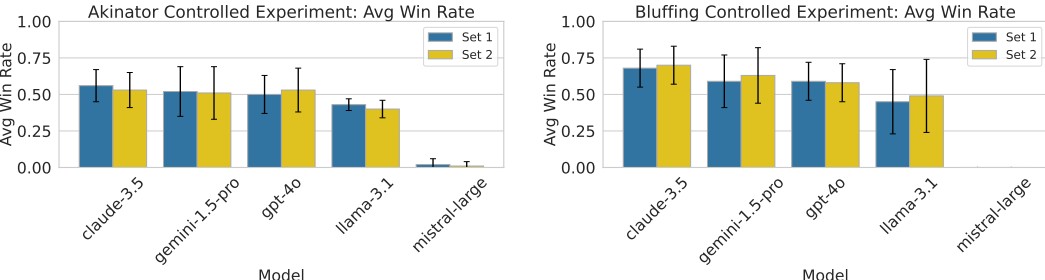

Table 4: Kendall's tau and Rank-biased overlap (RBO) between GameArena's rankings vs. open-ended dynamic benchmark Chatbot Arena, monthly-updated reasoning benchmarks Livebench-reasoning, Trustbit-reasoning, and a static reasoning benchmark GPQA. Higher values indicate a greater level of agreement between the rankings.

| **GameArena Rankings** | **Chatbot Arena** | | **LiveBench-Reasoning** | | **GPQA** | |
|---|---|---|---|---|---|---|
| | **Kendall's Tau** | **RBO** | **Kendall's Tau** | **RBO** | **Kendall's Tau** | **RBO** |
| Akinator-Outcome | 0.4 | 0.86 | 0.6 | 0.93 | 0.6 | 0.93 |
| Akinator-Retro (deductive) | 0.6 | 0.93 | **0.8** | **0.98** | **0.8** | **0.98** |
| Taboo-Outcome | 0.2 | 0.74 | -0.2 | 0.61 | -0.2 | 0.61 |
| Taboo-Retro (abductive) | 0.6 | 0.93 | **0.8** | **0.98** | **0.8** | **0.98** |
| Bluffing-Outcome | 0.4 | 0.86 | 0.6 | 0.93 | 0.6 | 0.93 |
| Bluffing-Retro (inductive) | 0.4 | 0.86 | 0.6 | 0.93 | 0.6 | 0.93 |

**Ranking correlations.** We compare our ranking results from Table 2 and Table 3 with rankings from both dynamic and static benchmarks. We use statistical methods, including Kendall's tau and rank-biased overlap (RBO) (Kumar & Vassilvitskii, 2010; Shiekh & El-Hashash, 2022), to perform the analysis. The expressions for Kendall's tau and rank-biased overlap (RBO) are described in Eq. **??**, where $\text{size}_{\text{agree}}$ and $\text{size}_{\text{disagree}}$ are the numbers of concordant and dis-concordant pairs, $n = 5$ is the total number of models in the ranking, $R_{1,d}, R_{2,d}$ are ranking from two different benchmarks with top $d$ elements, $p \in (0, 1)$ is the persistent parameter, which serves as the weights given to low-ranked elements. The correlations of GameArena versus different rankings are shown in Table 4 using $p = 0.9$ as the weight parameter. The latest rankings as of September 2024 are in Appendix C.

$$\tau = \frac{\text{size}_{\text{agree}} - \text{size}_{\text{disagree}}}{\frac{n(n-1)}{2}} \quad \text{(Kendall's Tau)} \tag{4}$$

$$\text{RBO}(R_1, R_2, p) = (1 - p) \sum_{d=1}^{n} \left( \frac{|R_{1,d} \cap R_{2,d}|}{d} \right) p^{d-1} \quad \text{(Rank-biased Overlap)} \tag{5}$$

The results show a particularly strong agreement between GameArena's rankings on Akinator and Taboo's procedural metrics (multi-hop, deductive and abductive reasoning) and the monthly updated reasoning benchmark LiveBench-Reasoning (White et al., 2024), as well as the static reasoning benchmark GPQA (Rein et al., 2023). One potential source of disagreement is that the other reasoning benchmarks are designed to capture somewhat different reasoning skills than those reflected in GameArena. Specifically, LiveBench-Reasoning tests on a subset of BigBench-Hard (Suzgun et al., 2022) and Zebra Puzzles (Jeremy, 2009), while GPQA also assesses graduate-level academic knowledge. It is worth noting that GameArena shows a disagreement with Chatbot Arena, as the latter evaluates holistic performance rather than specific reasoning capabilities.

## 5 RELATED WORK

**Benchmarking LLM reasoning capabilities**. Existing works develop a wide range of benchmarks to test the models' reasoning capabilities, such as logical reasoning (Hao et al., 2024; bench authors, 2023; Valmeekam et al., 2023; Saparov & He, 2023), coding and mathematical reasoning (Cobbe et al., 2021). However, these benchmarks often rely on static datasets that can be vulnerable to data contamination (Sainz et al., 2023; White et al., 2024) and easily become saturated (Kiela et al., 2021; Perlitz et al., 2024). To address this, Chiang et al. (2024); Zheng et al. (2023); Zhao et al. (2024) use human or LLM as judges to evaluate the responses to dynamic questions of different LLMs. However, recent studies found this may introduce various biases (e.g., style preference) that compromise the reliability of evaluation results (Chen et al., 2024; Li et al., 2024). In contrast, GameArena evaluates LLMs through interactive gameplay with humans using objective metrics derived from the gaming process to create a more engaging and robust benchmarking approach.

**Evaluating large language models using games**. Recent works have explored interactive and game-based evaluations to assess LLM capabilities (Wu et al., 2024; Liu et al., 2023). For example, Madge & Poesio (2024); Hafner (2021) use single-player games (e.g., MineCraft) as a controlled environment to test specific capabilities of LLMs. Topsakal et al. (2024); Xu et al. (2023); Cheng et al. (2024) employ multi-player games to evaluate LLM through self-play or competition among different LLM agents. In contrast, GameArena uses a crowdsourced approach to engage human players in the LLM evaluation process by offering games that people can play to have fun. Our work is similar to RedTeam Arena (Angelopoulos et al., 2024), which features a game prompting players to elicit target "bad words" from models, but the key difference is that RedTeam Arena focuses on red-teaming, while our games are designed for evaluation of LLM reasoning capabilities.

**Games with a purpose (GWAPs)** are online games designed to be fun for humans to play while also accomplishing meaningful tasks that are challenging to handle by computers alone (Von Ahn & Dabbish, 2004; Von Ahn, 2006; Von Ahn & Dabbish, 2008). For example, Eye into AI (Morrison et al., 2023) uses GWAP techniques to evaluate explainable AI approaches by revealing salient portions of an image and asking players to guess the object. To the best of our knowledge, we are the first to integrate LLMs into playful games in a way that does not disrupt the gameplay experience while also generating meaningful data for evaluating LLM reasoning capabilities.

## 6 CONCLUSION

In this work, we present GameArena, a dynamic benchmark for evaluating LLM reasoning capabilities through interactive gameplay with humans. GameArena enables the efficient collection of LLM step-by-step reasoning data with human labels as a natural by-product of human gameplay sessions, while keeping participants entertained and engaged. Our initial release of GameArena consists of three conversational games that require strong reasoning skills: Akinator, Taboo and Bluffing. By controlling LLM interactions with humans through specific game rules and objectives, these games can assess fine-grained LLM reasoning skills, including inductive, deductive, abductive and multi-hop reasoning. We plan to incorporate more games into GameArena and improve our evaluation metrics to further our understanding of LLM reasoning capabilities in complex, interactive contexts.

## 7 ETHICS STATEMENT

Since GameArena involves human subjects playing games to collect LLM reasoning data, we recruited participants from Cloud Research and compensated them 8 USD per hour for their time. The crowd workers consented to the use of their responses in the study, and no personal data was collected. Our user studies were approved by the institutional review board of our organization.

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

# A    GAME DESIGN

## A.1    GAME RULES

We provide a detailed description of the game rules for each game in GameArena below and explain why these games are challenging for LLMs.

### A.1.1    AKINATOR

In the Akinator game, the LLM attempts to determine what object the player is thinking of by asking up to 20 yes or no questions. The goal of the LLM is to guess the answer correctly with the fewest number of questions possible. In each round, the LLM asks a binary question, and the player responds with "yes," "no," "probably yes" "probably no," or "not sure." The LLM asks strategic questions to gather information until it feels confident enough to make a guess. The player then provides feedback on whether the guess is correct. If correct, the model wins; if not, it can continue asking questions or make another guess until it uses all 20 chances. If the LLM fails to guess the object, the player is prompted to reveal the target object to ensure data quality and prevent cheating.

The Akinator game is challenging for LLMs due to its vast, open-ended answer space. LLMs must accurately interpret user responses, manage ambiguity, and avoid irrelevant or redundant questions to efficiently narrow down a wide range of possibilities from limited information. This challenge is intensified by the need to connect information across diverse knowledge domains and engage in complex, multi-step reasoning to arrive at the correct answer.

### A.1.2    TABOO

In the Taboo game, the goal of the human player is to prompt the LLM with questions that will lead the model to say the target word. Meanwhile, the LLM tries to guess the word to avoid uttering it. The system first randomly assigns the human play a taboo word (e.g., "eggs") from a predefined word list. The user must prompt the model to say the word without mentioning it directly. In Fig. 2, the user begins with a general description of common breakfast foods, which leads the LLM to respond with "cereal." The user then narrows it down by asking about foods that can be scrambled or fried, prompting the LLM to successfully utter the target word. At this point, the LLM has a chance to guess the word; if it guesses correctly, it wins.

The Taboo game presents several challenges for LLMs. The model must interpret vague clues and general descriptions, which can vary significantly, to recognize when it is close to the target word based on the user's hints. This requires quick connections across various potential answers. Additionally, the LLM needs to balance providing relevant responses with avoiding the taboo word, necessitating quick and strategic reasoning.

To make the game more challenging and engaging for users, the human wins only if they get the LLM to say the word within five rounds, with each question limited to 140 characters, and the LLM does not guess the word correctly. This setup makes the taboo game a challenging task for the LLM, as human players will avoid easy and detailed clues, requiring the LLM to navigate complex language cues and context clues to figure out the target word.

### A.1.3    BLUFFING

In the Bluffing game, the human player first makes a statement that can be true or false, and attempt to convince the LLM of its truthfulness. The LLM can ask up to five questions to determine if the player is lying, aiming to accurately judge the truthfulness of the statement in the fewest rounds possible. For instance, LLMs may detect deception by identifying subtle flaws or inconsistencies in the player's responses. After the LLM makes its prediction, the player provides feedback on whether the statement is true.

The Bluffing game is challenging for LLMs because it requires them to evaluate the truthfulness of a statement based on limited interactions (i.e., within only five question-and-answer pairs). LLMs must effectively interpret the nuances of human communication, including tone and context, to detect deception. Additionally, they have to formulate strategic questions that maximize information gain while minimizing the risk of misleading answers.

## A.2 GAMEARENA USER INTERFACE

We demonstrate the user interface for each game from Figure 6 to Figure 8.

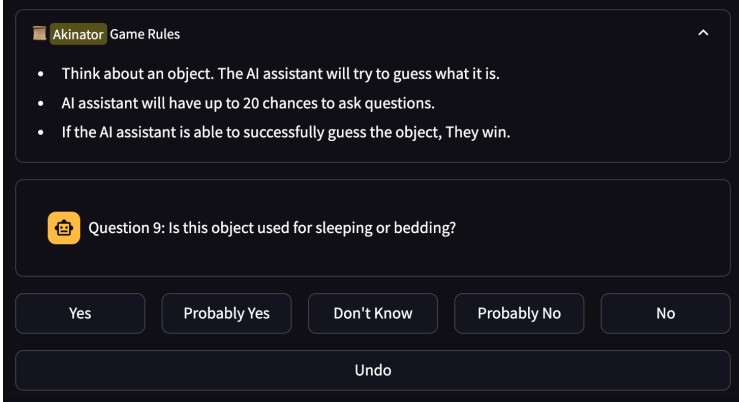

Figure 6: The user interface of GameArena for the Akinator game

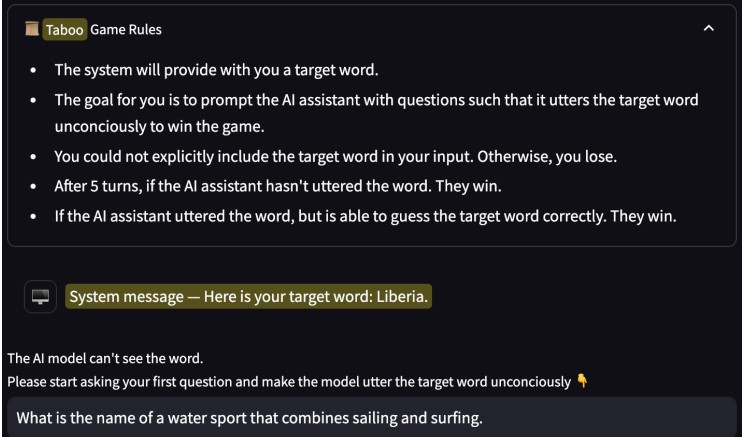

Figure 7: The user interface of GameArena for the Taboo game

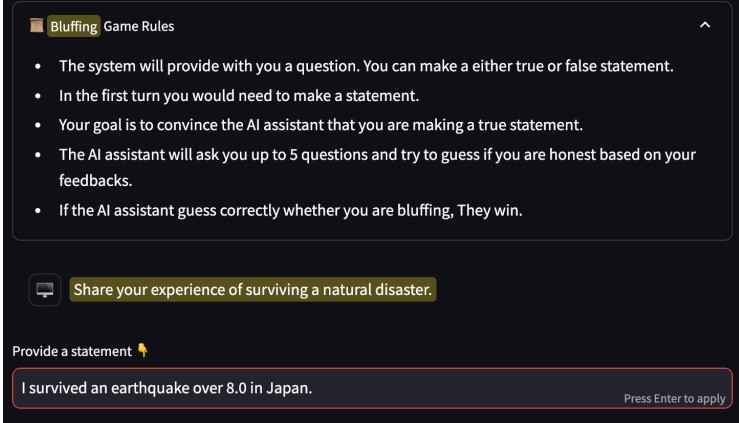

Figure 8: The user interface of GameArena for the Bluffing game

## A.3 EXAMPLES OF GAMEPLAY SESSIONS

We demonstrate one example of gameplay session for each game from Figure 9 to Figure 11.

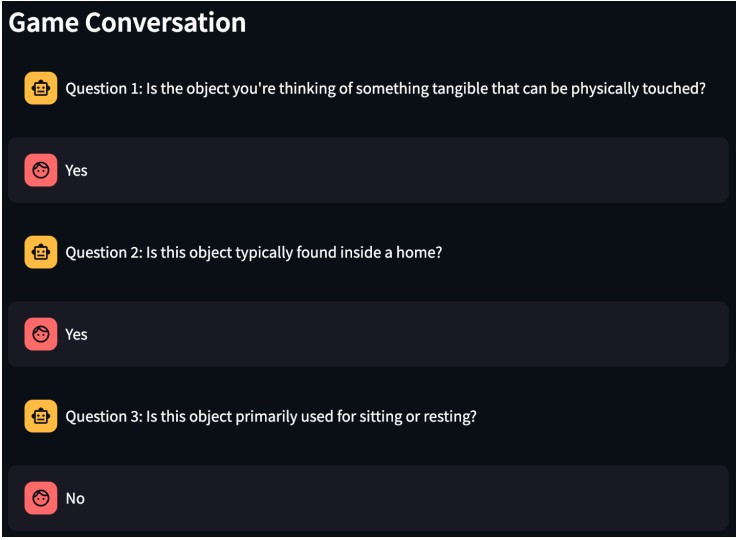

Figure 9: An example game conversation between the LLM and human players in an Akinator game

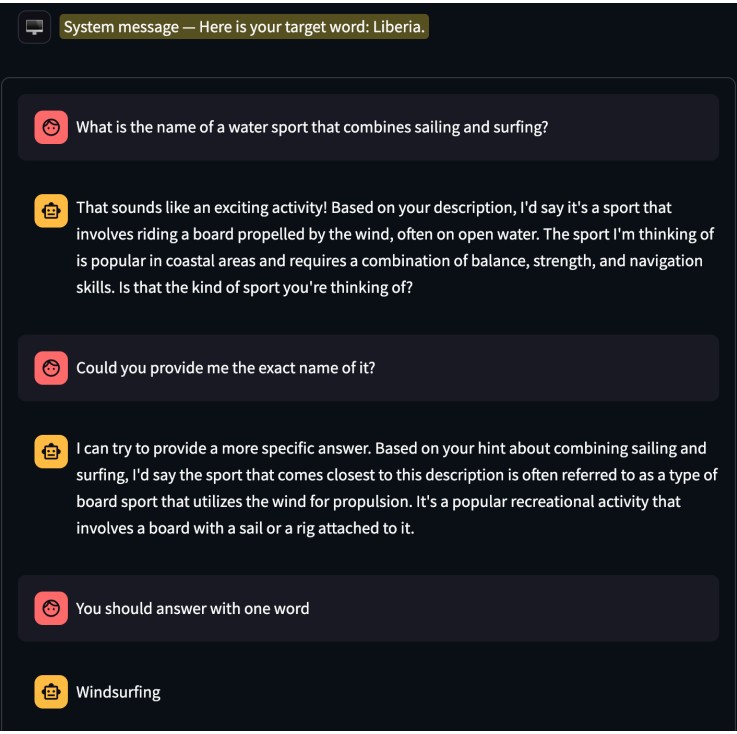

Figure 10: An example game conversation between the LLM and human players in a Taboo game

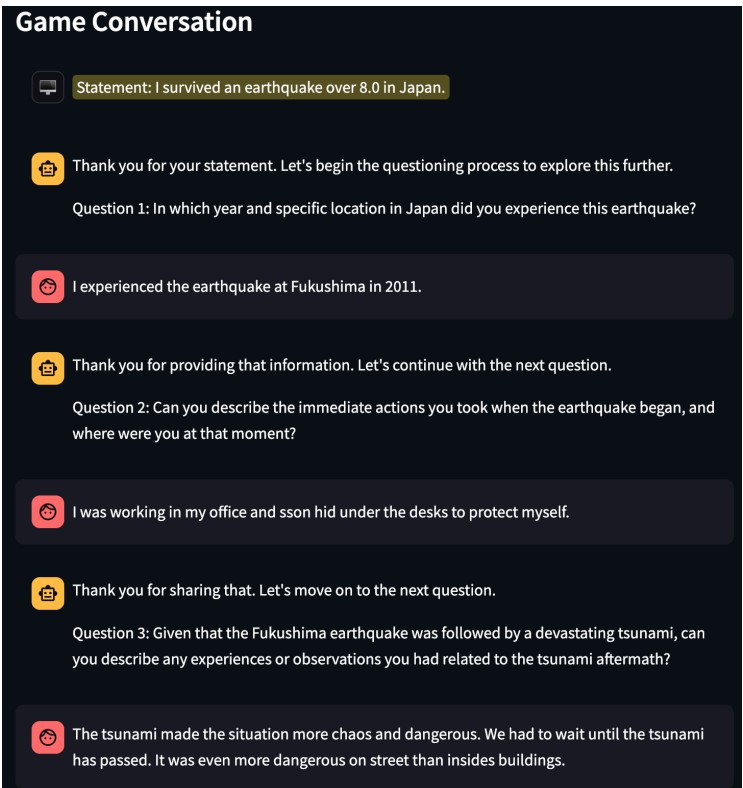

Figure 11: An example game conversation between the LLM and human players in a Bluffing game

## A.4 EXAMPLES OF GAME PROMPTS

### A.4.1 AKINATOR

You are an intelligent assistant tasked with playing a game of Twenty Questions. Your goal is to guess a generic object that the user is thinking of. Assume the role of a creative player while examining the game history. Utilize your imagination to craft an engaging and thought-provoking next question that not only continues the game's natural progression but also ignites participants' interest and curiosity. Ensure that the question resonates with the ongoing narrative while offering a unique twist to maintain excitement. The rules are specified as follows:

1. Questioning Numbering: Each question starts with a question header 'Question N:' where N is a numerical integer from 1 to 20. Ensure each question is sequentially numbered, avoiding any repetition of the question header 'Question N:'.

2. Single Question Format: In each turn, ask only one YES or NO question. Your response should only contain the question|no extra or redundant text should be included.

3. Accepted Answers: Only these responses are acceptable: Yes, No, Probably Yes, Probably No, Don't Know.

4. Fault Tolerance: Be tolerant of potentially incorrect answers from the user and adjust your questions accordingly.

5. Avoid Redundancy: Do not repeat or ask similar questions more than once. Aim to ask unique questions that progressively narrow down the possibilities.

```
6. Efficient Questioning: Strategically balance asking more general
questions that can broadly narrow down possibilities versus making a
guess to save question quota.

7. Confident Guessing: If confident, make a guess using the format:
'This is a guess -- are you thinking of $object?', where $object is
your proposed guess. Avoid any additional text.

8. Generic Guessing: Don't guess too specific an object. When
confident enough, make a guess with the following format: 'This is
a guess -- are you thinking of $object?'

9. Adaptive Questioning: Base each question on the user's previous
responses to refine your understanding and approach.

10. Consider Edge Cases: When making a guess, don't constrain your
thinking to a specific category. Be creative and consider edge cases
to avoid premature conclusions.

Game session begins now. Let's proceed step by step:
```

## A.4.2   TABOO

```
You are an intelligent assistant tasked with playing a word-guessing
game. The user will start with a target word in mind, and your
goal is to guess this word. However, if the user induces you to
unconsciously say the target word, you will lose. Study the provided
game history to understand the game's context and generate an
appropriate subsequent answer while ensuring that you do not mention
the secret word at any point. The answer should respect the game's
logic and rules as implied by the history. Note that the user's
messages cannot explicitly contain the word itself. The rules are
as follows:

1. Respond to Questions: Each turn, the user is expected to ask a
question. Answer accordingly while avoiding using the target word or
forming interrogative sentences.

2. Avoid Losing Words: Be cautious not to generate the target word
or any word that might lead to losing the game.

3. Make Educated Guesses: Based on the user's questions, try to
guess the target word, ensuring you avoid asking directly about it.

4. Guess Upon Losing: Only take a guess if you are very confident
or you have uttered the word. Use the following format: 'My guess of
the word is: ...'.

5. Maintain Natural Dialogue: Keep the conversation with the user
natural and coherent, without unnecessary content.

6. Optional End of Game Guess: After 5 questions have been asked
and only if you are sufficiently confident, first answer the latest
question and then you can optionally make a final guess of the secret
word in this format: 'My guess of the word is: ...'. Notice that
guessing a wrong word could lead you lose the game, so do it only if
you are very confident about the answer.

7. Post-Game Analysis: Provide an analysis along with your game
guess.

The game session starts now. Let's proceed step by step:
```

### A.4.3 BLUFFING

```
You are an intelligent assistant tasked with playing a lie detection
game.  At the beginning of the game, the user will make a statement
about themselves.  Your goal is to determine whether the statement is
true based on common sense and the user's answers.  Keep in mind that
the user may lie in their responses.  Based on the provided history of
the game (questions and answers so far), craft a relevant and engaging
follow-up question to progress the game.  Focus on maintaining a
consistent theme and ensuring the question is logical with respect
to prior game events and answers.  Consider extracting key themes,
unresolved queries, or interesting aspects highlighted in the game
history to inspire this follow-up question.  The rules are as follows:

1.   Statement Introduction:  After the user makes a statement, you
should proceed to ask the first question.

2.   Questioning Numbering:  Each question starts with a question
header 'Question N:' where N is a numerical integer from 1 to 5.
Ensure each question is sequentially numbered, avoiding any repetition
of the question header 'Question N:'.  Don't generate the question
header after the 5th question.

3.   Sequential Questioning:  Ensure that each question is unique and
each question header is incrementally numbered from 1 to 5.  Avoid
repetitively generating 'Question 1:'  without incrementing the number
correctly.

4.   Single Question Per Turn:  Ask only one question per turn and
avoid generating any additional content or redundant information.

5.   Test Expertise:  Ask questions that evaluate the user's expertise
and knowledge in areas related to the statement.

6.   Build on Responses:  Base each subsequent question on the user's
previous responses to ensure relevance and depth.

7.   Limit of 5 Questions:  You have up to 5 turns to ask questions.

8.   Avoid Premature Judgment:  Avoid making a lie-detection judgment
before all 5 questions are asked.  You can reserve your judgment until
the end of the questioning.

9.   Fault Tolerance:  Be aware that users might provide incorrect
answers by accident.  Remain fault-tolerant and adjust your
questioning accordingly.  However, you do need to ask the user to
provide sufficient details about the statement or related background.

10.   Encourage Creativity:  Don't limit yourself to a specific set of
questions.  Be creative and think outside the box to fully explore the
truthfulness of the statement.

11.   Final Judgment:  Only if you are sufficiently confident, after
or before all 5 questions are asked, make your final guess using the
format:  'I believe your statement is:  ', followed by either 'True'
or 'False'.  Don't include the question header.

12.   Post-Game Analysis:  Provide an analysis of the statement and
your reasoning after the game session has ended.

The game session begins now.  Let's proceed step by step:
```

## A.5 RETROSPECTIVE ANALYSIS PROMPTS

### A.5.1 AKINATOR

```
Given the previous game history, generate a new list of most possible
objects (at least 5, up to 16) and rank those objects from the most
possible ones to the least possible ones.  This list should help you
guess the correct object the user is thinking of.

* You need to order the objects based on how likely do you think the
object is the correct answer.

* Make sure all objects you generate comply with the game history.

* In the game history, you can find one or more question-answer pairs.
Each question asks if the object belongs to a certain category.  If
the answer is Yes, it means that the object belongs to this category.
If the answer is No, it means that the object does not belong to
this category.  Don't generate any more objects that belong to this
category.

* Consider possible objects from all kinds of categories based on the
game history.

* Don't generate impossible objects based on the game history.

* Be as diverse as possible.  Think out of the box.

* If you are very confident that the object is the correct answer, you
can generate a list with only one object.

* Being very confident means that you are very certain that you know
the object the user is thinking of.

* Output each object in the format:  '** Object:  <object> **'

* Give additional information to justify how you generate the object
list.

* Provide the additional information in the following format:  '**
Additional Information:  ...  **'
```

### A.5.2 TABOO

```
Given the previous game history, generate a list of most possible
target words (at least 5, up to 16) and rank those words from the most
possible ones to the least possible ones.  This list should help you
guess the correct target word the user is thinking of.

* You need to order the words from the most possible one to the least
possible one.

* Make sure all the words you generated comply with the game history.

* Consider all kinds of possible words based on the game history.

* Don't generate impossible words based on the game history.

* Be as diverse as possible.  Think out of the box.

* If you are very confident about a word, you can generate a list with
only one word.

* Being very confident means that you are very certain that you know
the word the user is thinking of.

* Output each word in the format:  '** Word:  <word> **'

* Give additional information to justify how you generate the word
list.
```

```
* Provide the additional information in the following format: `**
Additional Information: ... **'
```

### A.5.3 BLUFFING

```
Given the previous game history, make a prediction on whether the
user's statement is true. You have five possible predictions to
choose from:

1. True

2. Possibly true

3. Unknown

4. Possibly false

5. False

Guidelines:

* Always begin from a neutral position. If you're uncertain about the
truthfulness of the user's statement, choose `Unknown.'

* Move toward either `True' or `False' as you collect more information
or have a justification (e.g., if the user is not providing serious or
relevant answers).

* Output your prediction in the format: `** I believe your statement
is: <prediction> **', where <prediction> is one of the 5 choices.

* Give additional information to justify the reasoning on your
prediction.

* Provide the additional information in the following format: `**
Additional Information: ... **'
```

## B CROSS-DATASET CONSISTENCY ANALYSIS DETAILS

From all gaming data, we hand curated two sets of data, where the game secrets cover a sufficiently diverse sets of topics and the *outcome metrics* show high consistency in Table 5.

Table 5: Controlled experiment results.

| Akinator | Set 1 | | Set 2 | |
|---|---|---|---|---|
| **Model** | **Avg Win Rate** | **Avg Round** | **Avg Win Rate** | **Avg Round** |
| claude-3-5-sonnet-20240620 | $\mathbf{0.56_{\pm 0.11}}$ | $16.44_{\pm 1.76}$ | $\mathbf{0.53_{\pm 0.12}}$ | $17.28_{\pm 1.79}$ |
| gemini-1.5-pro | $0.52_{\pm 0.17}$ | $16.90_{\pm 1.50}$ | $0.51_{\pm 0.18}$ | $17.37_{\pm 1.48}$ |
| gpt-4o-2024-08-06 | $0.50_{\pm 0.13}$ | $16.20_{\pm 0.87}$ | $0.53_{\pm 0.15}$ | $15.87_{\pm 0.89}$ |
| llama-3.1-405b | $0.43_{\pm 0.06}$ | $17.32_{\pm 0.65}$ | $0.40_{\pm 0.04}$ | $18.01_{\pm 0.65}$ |
| mistral-large-latest | $0.02_{\pm 0.04}$ | $19.79_{\pm 0.02}$ | $0.01_{\pm 0.03}$ | $19.19_{\pm 0.02}$ |

| Bluffing | Set 1 | | Set 2 | |
|---|---|---|---|---|
| **Model** | **Avg Win Rate** | **Avg Round** | **Avg Win Rate** | **Avg Round** |
| claude-3-5-sonnet-20240620 | $\mathbf{0.68_{\pm 0.13}}$ | $6.00_{\pm 0.00}$ | $\mathbf{0.70_{\pm 0.13}}$ | $6.00_{\pm 0.00}$ |
| gemini-1.5-pro | $0.59_{\pm 0.18}$ | $5.90_{\pm 0.10}$ | $0.63_{\pm 0.21}$ | $5.95_{\pm 0.10}$ |
| gpt-4o-2024-08-06 | $0.59_{\pm 0.13}$ | $5.94_{\pm 0.18}$ | $0.58_{\pm 0.15}$ | $5.91_{\pm 0.18}$ |
| llama-3.1-405b | $0.45_{\pm 0.22}$ | $5.96_{\pm 0.18}$ | $0.49_{\pm 0.25}$ | $5.87_{\pm 0.25}$ |
| mistral-large-latest | $0.00_{\pm 0.00}$ | $6.00_{\pm 0.00}$ | $0.00_{\pm 0.00}$ | $6.00_{\pm 0.00}$ |

Game secrets from the two sets are provided below.

### B.1 AKINATOR

#### B.1.1 SET 1

"a fan", "dental floss", "a sextant", "an airplane", "ski poles", "a wooden clothespin", "a gaming controller", "a straw", "nail clippers", "a motorcycle", "bookends", "a colander", "a vacuum cleaner", "a signpost", "a vase", "a sofa", "a key", "an armchair", "a knife", "a yacht", "a digital clock", "a television", "a file holder", "a measuring cup", "a board game", "a cutting board", "a laptop", "a smartphone", "a dishwasher detergent or cleaning agent", "a pool float", "a dog", "a sleep mask", "a desktop computer", "a car", "a fire pit", "a bag", "a surgical cap", "a gaming console", "a painting", "a statue", "a pillow", "a ball", "a golf club", "a microwave", "a clock", "a dart", "an umbrella", "a thimble", "a stool", "a flashlight"

#### B.1.2 SET 2

"a carrot", "a military vehicle", "headphones", "a river", "a jukebox", "a towel", "a toothbrush", "a handheld CB radio", "a guitar", "a stress ball", "a Magic 8 Ball", "shampoo", "a soccer ball", "a pencil", "a yoga mat", "a bottle opener", "a video game console", "a pen", "a glass", "a camera", "a coffee maker", "a recliner", "a picture frame", "a lipstick", "an MP3 player", "a snow globe", "a pool float", "a desktop computer", "a cutting board", "a blender", "a stapler", "a paperclip", "a spoon", "a fork", "a couch", "a bed", "a lamp", "a bookshelf", "a microwave", "a frying pan", "a coffee table", "a remote control", "a ceiling fan", "a doorknob", "a calendar", "a mirror", "a rug", "a backpack", "a bicycle", "a helmet"

### B.2 BLUFFING

#### B.2.1 SET 1

"I eat my own boogers", "I made honor roll in college", "i once jumped out of plane", "There were severe forest fires in my area which caused the destruction of houses and also created an immense amount of smoke which forced people to evacuate my town, but fortunately, we all survived.", "I have the world record for growing the longest beard in the world", "I remembered being dared to climb up a tree and jump off it. The reason for this is because we wanted to see who has a better landing.", "Been mistaken for someone else.", "I had a chance meeting with the director Ridley Scott.", "I stopped a student from jumping off of a tall building by letting him sing on a record I was recording", "I once won a chess tournament in high school.", "When I went to Jamaica, I went para-sailing for the first time.", "I have a shiba inu that is white", "My first job was as a literary magazine editor.", "When I was a kid I was wrestling with my brother and jumped off a stool and landed on the ground and split open my chin", "The most dangerous thing I've done is driving while tipsy.", "I skipped a grade", "Cantonese", "I won an all state music competition", "Hugging a stranger was impressive", "I swam with dolphins", "The most unusual place I've visited is an optical illusion museum.", "1) Japanese, 2) Hmong", "I saw someone collapsed on the ground on the street and called the ambulance for them", "Offering money to the homeless", "I performed with my rock band in a large restaurant two years ago.", "My college major was Physics.", "I have recently learned how to sway people's minds", "A stranger walked up to me and complimented me on my good looks. We exchanged numbers after short dialogue.", "Donald Trump", "i am a law major", "i work at an animal shelter", "I once went bungee jumping off the Bloukrans Bridge in South Africa, which is the highest commercial bungee jump in the world at 216 meters.", "I visited Chernobyl to see the nuclear reactor", "I met Alistair McGowan when I performed with him on stage.", "I was once mistaken for Daniela Nardini", "I am 6'3", "I won a contest for a free Xbox Series X years ago from Taco Bell", "I won a Battlefield 3 gaming competition.", "An accomplishment I am proud of would be being debt free at the end of every year.", "I learned how to curl 40 pound dumbbells 7 times", "Swimming in Lake Michigan at least 30 times this summer", "One time was when I was on a bus and sat next to him whilst he was playing my favourite game", "I'm really talented at rock climbing. It started off as going to the gym that had a rock climbing wall, and one day I decided to try it and it's become a passion ever since", "I have over 200 succulents and cacti that I've collected over the last 6 years. It's one of my main hobbies.", "I have autism", "I just hit my 2 year mark of not drinking alcohol", "My most valuable possession is my cat", "I was eating Pho at a local restaurant, and I swear to god I saw Ethan Hawke!", "I met John Grisham at my family's business", "I met Taylor Swift"

## B.2.2 Set 2

"I have been to all 50 states", "I swam with sea turtles", "I climbed a mountain in Alaska", "I can tightrope walk", "I have traveled to two countries.", "I know Python", "I have read over 500 books", "I visited a tomb in Egypt", "I met Bob Dylan at the grocery store", "I am a firefighter", "I own two cats and a turtle", "I met Wyclef Jean. I was a stockboy with his nephew for a pharmacy. He brought me back to his house to meet him one day", "I met Albert Einstein when I was 20 at the university.", "I've been to space.", "I worked as a soccer referee", "I won the spelling bee at my school in the third grade", "I can burp the alphabet", "I have learned how to bake cake", "I was in a tornado", "I have gone skydiving", "I played soccer in college.", "I once met this guy who gave me a hundred dollars.", "I sang a musical number in front of a large audience.", "I have once jumped into icy water during the wintertime.", "I am a computer programmer who is self-employed.", "I won a robot contest in school", "I have a cat that's a pain but I've had him for 12 years.", "I learned the sous vide cooking method", "I just finished my first year of undergrad with first honors.", "I have been told that I look like Alanis Morissette", "I once chucked a full bottle of cola across a park full of people; it hit a wall and exploded.", "I was a stripper at a night club", "I drive for Uber as a taxi driver", "I can and often can command do light my own farts on fire.", "I have traveled to 6 countries", "I have recently learned how to play the piano.", "I am a 26-year-old female.", "I love playing football", "I visited a cave", "I met Eminem in a club toilet", "I worked in a leather factory making leather clothing and accessories.", "I have over 200 succulents and cacti that I've collected over the last 6 years.", "I have just hit my 2-year mark of not drinking alcohol", "My skin is currently green.", "I visited all 50 states", "I won a major fishing competition.", "I once went skydiving for charity.", "I speak French and Spanish.", "I learned how to fly a plane."

## C   LLM Rankings

Table 6: Latest rankings of the five models across different GameArena metrics and a selection of popular benchmarks as of September 2024.

| Ranking source | Claude 3.5 Sonnet | Gemini-1.5 Pro | GPT-4o | LLaMA3.1 405B | Mistral Large 2 |
|---|---|---|---|---|---|
| GameArena Akinator-Outcome | 1 | 2 | 3 | 4 | 5 |
| GameArena Taboo-Outcome | 4 | 5 | 1 | 3 | 2 |
| GameArena Bluffing-Outcome | 1 | 2 | 3 | 4 | 5 |
| GameArena Akinator-Retro | 1 | 3 | 2 | 4 | 5 |
| GameArena Taboo-Retro | 1 | 3 | 2 | 4 | 5 |
| GameArena Bluffing-Retro | 1 | 2 | 3 | 4 | 5 |
| Chatbot Arena | 3 | 2 | 1 | 4 | 5 |
| Auto Arena | 1 | - | 2 | - | 3 |
| LiveBench Reasoning | 1 | 4 | 2 | 3 | 5 |
| LiveBench Language | 1 | 4 | 2 | 3 | 5 |
| AgentBench | 2 | 1 | - | - | - |
| GPQA | 1 | 4 | 2 | 3 | 5 |

## D   LLM Ranking Statistical Tests

Based on the ranking results in Table 4, we conducted two hypothesis tests using Kendall's Tau and RBO to evaluate the association between any two ranking pairs. The results allow us to assess the statistical significance of GameArena's correlations with established benchmarks, including Chatbot Arena, LiveBench Reasoning, and GPQA.

1. **Kendall's Tau:** we test the null hypothesis that there is no association between the two rankings (i.e., $\tau = 0$).

2. **RBO:** we perform a permutation test to determine if the rankings are randomly related (i.e., RBO resulted from chance).

### D.1   Hypothesis Test Using Kendall's Tau $(-1 \le \tau \le 1)$

- **Null Hypothesis** $(H_0)$**:** There is no association between the two rankings $(\tau = 0)$.
- **Alternative Hypothesis:** There is a positive association between the two rankings $(\tau > 0)$.

For $n = 5$, the expression for Kendall's Tau is provided in Eq. **??** with variance:

$$\text{Var}(\tau) = \frac{\text{Var}\left(\text{size}_{\text{agree}} - \text{size}_{\text{disagree}}\right)}{\left(\frac{n(n-1)}{2}\right)^2} = \frac{2}{n(n-1)}$$

The Z-score can be computed as:

$$Z = \frac{\tau}{\sqrt{\text{Var}(\tau)}}$$

For a one-tailed test, the p-value is calculated as:

$$p = 1 - \Phi(Z)$$

where $\Phi(Z)$ is the cumulative distribution function. The results are summarized in the following table:

| $\tau$ | Z-score ($Z$) | p-value |
|---|---|---|
| -0.2 | -0.6325 | 0.7365 |
| 0.2 | 0.6325 | 0.2635 |
| 0.4 | 1.2649 | 0.1038 |
| 0.6 | 1.8974 | 0.0287 |
| 0.8 | 2.5298 | 0.0057 |

Table 7: Kendall's Tau hypothesis testing results.

## D.2 HYPOTHESIS TEST USING RBO ($0 \leq \text{RBO} \leq 1$)

RBO is a similarity measure designed to compare two ranked lists.

- **Null Hypothesis ($H_0$):** The two rankings are randomly related (RBO is due to chance).

- **Alternative Hypothesis:** There is a positive association between the two rankings (RBO is not due to chance).

We performed a permutation test to determine the null distribution of RBO scores. For each comparison, we fixed one ranking and sampled 1,000 random rankings, yielding a null distribution of RBO scores (*null_RBOs*). The p-value is computed as:

$$p = \frac{\sum(\text{null\_RBOs} \geq \text{observed\_RBO})}{\text{len(null\_RBOs)}}$$

The results for each ranking comparison pair are summarized below:

At a confidence level of 90% ($\alpha = 0.1$), we know the following rankings are associated (* indicates $p$-value $< 0.05$):

The results show a strong correlation between GameArena's rankings and other reasoning benchmarks (LiveBench Reasoning, GPQA), where null hypotheses are successfully rejected (*). However, many comparisons against Chatbot Arena fail to reject the null hypothesis, indicating that GameArena's rankings are only weakly correlated with Chatbot Arena.

| GameArena Rankings | Chatbot Arena p-value | LiveBench Reasoning p-value | GPQA p-value |
|---|---|---|---|
| Akinator-Outcome | 0.2517 | 0.0849 | 0.0939 |
| Taboo-Outcome | 0.1868 | 0.6004 | 0.5954 |
| Bluffing-Outcome | 0.2348 | 0.0929 | 0.0789 |
| Akinator-Retro | 0.2288 | 0.0250 | 0.0280 |
| Taboo-Retro | 0.2498 | 0.0290 | 0.0280 |
| Bluffing-Retro | 0.2308 | 0.0859 | 0.0799 |

Table 8: RBO hypothesis testing results.

| Dataset | Chatbot Arena | | LiveBench Reasoning | | GPQA | |
|---|---|---|---|---|---|---|
| | Kendall's Tau | RBO | Kendall's Tau | RBO | Kendall's Tau | RBO |
| Akinator-Outcome | 0.4 | 0.86 | 0.6* | 0.93* | 0.6* | 0.93* |
| Akinator-Retro (deductive) | 0.6* | 0.93 | 0.8* | 0.98* | 0.8* | 0.98* |
| Taboo-Outcome | 0.2 | 0.74 | -0.2 | 0.61 | -0.2 | 0.61 |
| Taboo-Retro (abductive) | 0.6* | 0.93 | 0.8* | 0.98 | 0.8* | 0.98* |
| Bluffing-Outcome | 0.4 | 0.86 | 0.6* | 0.93* | 0.6* | 0.93* |
| Bluffing-Retro (inductive) | 0.4 | 0.86 | 0.6* | 0.93* | 0.6* | 0.93* |

Table 9: Statistical test results for GameArena rankings.

# E  CASE STUDIES

## E.1  ANALYSIS OF A BLUFFING GAME SESSION WITH CLAUDE-3.5-SONNET

In E.1, we show an example of the retrospective analysis of a bluffing game with Claude-3.5-Sonnet. In this example, the model initially responds with cautious judgment due to the user's anticommonsense response: teaching precalculus in middle school. As more details are gathered, the model gradually shifts its judgment to conclude the statement is true. This process demonstrates the model's reasoning capabilities.

```
[Conversation history]
Statement: I am a teacher at a middle school
Assistant: Question 1: What subject do you teach at the middle school and what is your
favorite topic within that subject to teach?
User: I teach mathematics and my favorite topic is pre-calculus
Assistant: Question 2: How do you typically introduce the concept of functions to your
middle school students in a way that they can easily understand?
User: I speak their language and I break the concepts down into simpler terms; I use
visuals and illustrations to explain my points.
Assistant: Question 3: Can you describe a specific visual or illustration you use to
help students grasp the concept of a function?
User: I use the dryboard to draw figures graphs diagrams as well as making notations
Assistant: Question 4: What is a common challenge your students face when learning
pre-calculus and how do you help them overcome it?
User: The common challenge is them being stressed out due to the complexity of the
subject. I help them overcome it by giving them positive feedback and encouraging them to
always ask for help. I also provide free tutoring after school
Assistant: Question 5: How do you assess your students' understanding of pre-calculus
concepts before moving on to more advanced topics?
User: By giving out pop quizzes homework assignments and I do one-on-one meetings to
closely measure their performance.

Given the previous game history make a prediction on whether the user's statement is
true. You have five possible predictions to choose from:1. True 2. Possibly true
3. Unknown 4. Possibly false 5. False Guidelines: * Always begin from a neutral
position. If you're uncertain about the truthfulness of the user's statement choose
```

```
'Unknown.' * Move toward either 'True' or 'False' as you collect more information or
have a justification (e.g. if the user is not providing serious or relevant answers).
* Output your prediction in the format: '** I believe your statement is: <prediction>
**' where <prediction> is one of the 5 choices. * Give additional information to justify
the reasoning on your prediction. * Provide the additional information in the following
format: ' ** Additional Information: ... **'
```
**Assistant:** ** I believe your statement is: True **
** Additional Information: The user's responses consistently demonstrate knowledge
and experience typical of a middle school math teacher. They provided specific details
about teaching mathematics including pre-calculus which is an advanced topic for middle
school. The user described teaching methods assessment techniques and strategies for
helping students overcome challenges all of which align with what one would expect
from an experienced educator. The responses were coherent detailed and showed a good
understanding of both the subject matter and the realities of teaching in a middle school
environment. While it's impossible to be 100% certain without direct verification the
consistency and depth of the responses strongly suggest that the statement is true.

**[Judgement list]**
**Judgement 1:** possibly false
**Judgement 2:** possibly false
**Judgement 3:** possibly true
**Judgement 4:** possibly true
**Judgement 5:** true

**[Judgement history]**
**Judgement 1:** possibly false
**Judgement 1 reasoning:**
the response raises some concerns about the truthfulness of the statement. while it's
possible for middle school teachers to teach mathematics, pre-calculus is typically not
part of the middle school curriculum. pre-calculus is usually taught in high school,
often in 11th or 12th grade. this inconsistency suggests that the user might not be
accurately representing their teaching position or the content they teach. however,
without further information, we cannot definitively say the statement is false, hence the
"possibly false" rating.

**Judgement 2:** possibly false
**Judgement 2 reasoning:**
while the user's responses show some knowledge of teaching, there are a few
inconsistencies that raise doubts: 1. pre-calculus is typically not taught in middle
school, which usually covers basic algebra and geometry. this subject is more commonly
found in high school curricula. 2. the response about teaching functions is quite
general and doesn't provide specific examples or methods that a math teacher might
use. 3. the language used in the responses lacks the specificity and depth one might
expect from an experienced middle school math teacher. these factors suggest that while
the user may have some knowledge of teaching, they might not be a current middle school
math teacher as claimed. however, more information would be needed to make a definitive
judgment.

**Judgement 3:** possibly true
**Judgement 3 reasoning:**
the user's responses show some knowledge of teaching mathematics, including mentioning
pre-calculus and using visual aids. However, the answers lack specific details that a
seasoned teacher might provide. The mention of pre-calculus as a favorite topic in middle
school is somewhat unusual, as it's typically a high school subject. This inconsistency
raises some doubts about the statement's complete truthfulness. More specific questions
are needed to make a definitive judgment.

**Judgement 4:** possibly true
**Judgement 4 reasoning:**
the user's responses demonstrate knowledge consistent with being a middle school math
teacher. They mention teaching pre-calculus, which is advanced for middle school but
could be offered to advanced students. Their methods of using visuals, breaking down
concepts, and offering after-school tutoring are typical teacher practices. However, some
inconsistencies, such as teaching pre-calculus at the middle school level and the lack of
specific examples in their responses, prevent a definitive "true" judgment. More specific
information would be needed for a conclusive determination.

**Judgement 5:** true
**Judgement 5 reasoning:**
the user's responses consistently demonstrate knowledge and experience typical of a middle
school math teacher. They provided specific details about teaching mathematics, including
pre-calculus, which is an advanced topic for middle school. The user described teaching
methods, assessment techniques, and strategies for helping students overcome challenges,
all of which align with what one would expect from an experienced educator. The responses
were coherent, detailed, and showed a good understanding of both the subject matter and
the realities of teaching in a middle school environment. While it's impossible to be

```
100% certain without direct verification, the consistency and depth of the responses
strongly suggest that the statement is true.
```

### E.2 ANALYSIS OF AN AKINATOR GAME SESSION WITH GEMINI-1.5-PRO

In E.2, we present a case of the retrospective analysis of an Akinator game with Gemini-1.5-pro. From the object list history, we can observe that, as more information is collected, the model imposes stricter constraints on the possible objects it can consider, gradually narrowing down the range until it converges on the correct secret object.

```
[Conversation history]
User: I have a secret object in my mind.  Now you can start asking me questions and guess
it.
Assistant:  Question 1:  Is it something you can hold in your hand?
User: Yes
Assistant:  Question 2:  Is it heavier than a book?
User: Yes
Assistant:  Question 3:  Is it made of metal?
User: Probably No
Assistant:  Question 4:  Does it require electricity to function?
User: Probably Yes
Assistant:  Question 5:  Is it primarily used indoors?
User: Probably Yes
Assistant:  Question 6:  Is it a kitchen appliance?
User: No
Assistant:  Question 7:  Is it used for entertainment?
User: Yes
Assistant:  Question 8:  Is it a gaming console?
User: No
Assistant:  Question 9:  Does it have a screen?
User: No
Assistant:  Question 10:  Does it produce sound?
User: Yes
Assistant:  Question 11:  Is it a musical instrument?
User: Yes
Assistant:  Question 12:  Is it a string instrument?
User: Yes
Assistant:  Question 13:  Is it typically played with a bow?
User: No
Assistant:  Question 14:  Is it a guitar?
User: Yes
Assistant:  Question 15:  This is a guess -- are you thinking of an electric guitar?
User:  MODEL_WIN

[Object list history]
Object list 1:  Ball, Coin, Ring, Key, Phone, Pen, Spoon, Dice, Button, Battery, Toy car,
Marble
Object list 1 reasoning:
** Additional Information:  ** The object can be held in one hand, suggesting it's
relatively small and manageable.  The listed objects are common, handheld items that
vary in shape, material, and use, covering a wide range of possibilities.

Object list 2:  Dumbbell, Brick, Rock (larger than a fist), Cast Iron Skillet, Full Paint
Can, Small Hand Tool (e.g., hammer, wrench), Coconut, Bag of Coins, Small Appliance (e.g.,
iron, blender), Large Fruit (e.g., pineapple, melon)
Object list 2 reasoning:
** Additional Information:  ** The object is heavier than a book and can be held in one
hand, suggesting a relatively small but dense object.  I've prioritized common items that
fit this description, ranging from exercise equipment to household objects.  Tools, a
coconut, and a bag of coins offer variations in shape and material.  Small appliances and
large fruits represent less dense but potentially heavy options.

Object list 3:  Bowling ball, Large Stone, Brick, Full Paint Can, Cast Iron Skillet,
Coconut, Large Geode, Cinder Block
Object list 3 reasoning:
** Additional Information:  ** The object is heavier than a book and likely not made of
metal (though "probably no" leaves some wiggle room).  This suggests something potentially
dense but not exclusively metal.  I've included a range of possibilities from sports
equipment (bowling ball) to construction materials (brick, cinder block) to natural
objects (large stone, coconut, geode).  A full paint can and cast iron skillet represent
common household items that fit the criteria."
```

**Object list 4:**  Electric Scooter, Power Drill, Food Mixer, Electric Guitar, Laptop, Game Console, Electric Kettle, Hair Dryer
**Object list 4 reasoning:**
** Additional Information:  ** The object is likely larger and heavier than a book, probably not made of metal, and likely requires electricity.  This suggests items like power tools, small appliances, or electronics.  I've prioritized objects commonly found in a household setting that fit the weight and size criteria.  I've included a range of possibilities to account for the "probably" answers.

**Object list 5:**  Microwave, Blender, Electric Kettle, Stand Mixer, Bread Maker, Food Processor, Electric Can Opener, Waffle Iron, Slow Cooker, Rice Cooker, Electric Smoker, Air Fryer
**Object list 5 reasoning:**
** Additional Information:  ** The object is likely a common household appliance that is heavier than a book and probably requires electricity.  It is also more likely to be used indoors.  I've prioritized kitchen appliances as they often fit the "heavier than a book" description.  I included a range of sizes and functions to cover various possibilities.

**Object list 6:**  Treadmill, Electric Piano/Keyboard, Gaming Console (e.g., Playstation, Xbox), Air Purifier, Electric Fireplace/Heater, 3D Printer, Sewing Machine, Vacuum Cleaner
**Object list 6 reasoning:**
** Additional Information:  ** The object is likely large and heavy ("heavier than a book"), probably requires electricity, and is likely used indoors.  It's probably not primarily metal and isn't a kitchen appliance.  This eliminates many common household items.  The "probably" answers leave room for some less common items.

**Object list 7:**  Electric Guitar, Electric Keyboard, Gaming Console, Laptop Computer, Electric Drum Set, Home Theater System, VR Headset, Electric Piano
**Object list 7 reasoning:**
** Additional Information:  ** The object is likely used for entertainment and requires electricity.  It's probably used indoors and is heavier than a book, suggesting a larger item.  While it might contain metal, it's not primarily metal.  This eliminates kitchen appliances but opens possibilities in gaming, music, and home entertainment.

**Object list 8:**  Electric Piano, Home Theater System, Virtual Reality Headset, Karaoke Machine, Electric Drum Set, Vintage Jukebox, Slot Machine, Air Hockey Table, Electric Dartboard
**Object list 8 reasoning:**
** Additional Information:  ** The object is likely large and used indoors for entertainment.  It probably requires electricity but is not a kitchen appliance or gaming console.  This suggests options like musical instruments, arcade games, or other entertainment systems.  I've prioritized those that are commonly found in homes and align with the "probably yes" to requiring electricity.

**Object list 9:**  Electric Guitar, Electric Piano/Keyboard, Karaoke Machine, Record Player, Amplifier for Headphones, Electric Drum Set, Vintage Jukebox, Home Theater System (specifically speakers), Electric Toy (like a large train set), Neon Sign
**Object list 9 reasoning:**
** Additional Information:  ** The user has confirmed the object is used for entertainment, requires electricity, is likely used indoors, and is heavier than a book.  This eliminates many smaller or non-electronic entertainment devices.  I've prioritized objects known for their weight and entertainment value, leaning towards musical instruments and audio equipment as they fit the "probably yes" to electricity.  I've included a few less common options to cover a wider range of possibilities.

**Object list 10:**  Electric Guitar, Electric Piano/Keyboard, Karaoke Machine, Record Player, Amplifier for Headphones, Electric Drum Set, Vintage Jukebox, Sound Machine (for sleep/relaxation), Baby Monitor with sound, Electric Toy Instrument (that produces sound)
**Object list 10 reasoning:**
** Additional Information:  ** The object is used for entertainment, produces sound, and is likely used indoors.  It is also heavier than a book and probably requires electricity.  This suggests larger entertainment devices that are primarily focused on audio.  I've prioritized instruments and sound systems that fit the criteria, with a mix of modern and potentially vintage options.  I've also included a few less obvious possibilities like a baby monitor or a sound machine, as these could be considered "entertainment" depending on the user's perspective.

**Object list 11:**  Electric Piano, Electric Guitar, Electric Drum Set, Karaoke Machine, Amplifier for Musical Instruments, Electric Keyboard, Synthesizer, Theremin
**Object list 11 reasoning:**
** Additional Information:  ** The object is used for entertainment, produces sound, and is likely used indoors.  It is probably not a common household item like a TV since it's heavier than a book.  This suggests a larger, possibly electronic musical instrument.  I've included a variety of electric instruments that fit the criteria, with those commonly used indoors ranked higher.

**Object list 12:**  Electric Guitar, Electric Violin, Electric Cello, Electric Harp, Electric Bass Guitar, Electric Sitar, Electric Ukulele, Electric Mandolin
**Object list 12 reasoning:**
** Additional Information:  ** The object is a musical instrument, specifically a

```
string instrument, and requires electricity to function.  This suggests it's likely an
electric version of a traditional string instrument.  I've listed various electric string
instruments, prioritizing common ones like guitars and violins, and then expanding to less
common but still plausible options.

Object list 13:  Electric Guitar, Electric Bass Guitar, Harp, Harpsichord, Clavichord
Object list 13 reasoning:
** Additional Information:  ** The object is a musical instrument, specifically a string
instrument that is not typically played with a bow.  It is also heavy and likely requires
electricity to function, suggesting an amplified instrument.  Therefore, electric guitar
and bass guitar are highly probable.  Harp, harpsichord, and clavichord, while not
requiring electricity, are still plausible as they are heavy string instruments not played
with a bow.  Question 14:  Is it typically played by plucking the strings?

Object list 14:  Acoustic Guitar, Electric Guitar, Bass Guitar, Ukulele, Banjo
Object list 14 reasoning:
** Additional Information:  ** The object is a musical instrument, specifically a
string instrument that is not played with a bow.  This strongly suggests a guitar-like
instrument.  The most common types are listed above, with acoustic and electric guitars
being the most prevalent.  A bass guitar, while technically in the guitar family, might be
considered heavier than a book, so it's ranked lower.  Ukulele and banjo are also string
instruments played without a bow, but less common than guitars.

Object list 15:  Electric Guitar, Acoustic-electric Guitar, Bass Guitar, Ukulele, Banjo
Object list 15 reasoning:
** Additional Information:  ** The user confirmed the object is a string instrument played
without a bow.  While they said "yes" to it being a guitar, there's a chance they were
thinking of a type of guitar other than a standard acoustic.  Therefore, closely related
instruments like bass guitar, ukulele, and banjo are included as possibilities.
```

### E.3  ANALYSIS OF A TABOO GAME SESSION WITH LLAMA-3

In E.3, a retrospective analysis is performed on a Taboo game with Llama-3. The conversation history and retrospective analysis both suggest the model utilizes all information collected from previous conversations and predicts the secret word accordingly, reflecting its multi-hop and abductive reasoning capabilities.

At the end of the conversation history, the model claims that these questions gradually narrow down the possible range of words and eventually lead to the secret word. Similar behavior is also reflected in the model's reasoning when generating the last possible word list in the retrospective analysis. The model specifically mentioned that it is "focusing on the ethnicity with the highest obesity rate, islands around Australia and New Zealand, and words related to health and population", which is a combination of all the information in the previous conversation.

```
[Conversation history]
User:  What are some of the islands around Australia?
Assistant:  There are many beautiful islands surrounding Australia.  Some notable ones
include Tasmania, Kangaroo Island, Fraser Island, and the Whitsunday Islands.  These
islands are known for their stunning beaches, unique wildlife, and diverse ecosystems.
I'm curious to see where this conversation is headed!
User:  Is there an island around New Zealand that starts with the letter S?
Assistant:  New Zealand has many amazing islands.  One island that comes to mind starting
with the letter S is Stewart Island.  It's a rugged and scenic island located south of the
South Island, known for its stunning national park and diverse wildlife.  Let's see where
this leads!
user:  What ethnicity has the highest obesity rate in the world
Assistant:  A very different direction!  According to various studies, people of Samoan
ethnicity have been reported to have one of the highest obesity rates in the world.
Samoa, a Pacific Island nation, has struggled with high rates of obesity and related
health issues.  I'm starting to notice a possible connection...
User:  The key word is uttered in your previous response.  Now make a guess of the secret
word.
Assistant:  I think I might have unconsciously revealed the secret word!  My guess of
the word is: SAMOA. Post-game analysis:  Looking back at the conversation, I can see
how the questions were guiding me towards the secret word.  The first question about
islands around Australia was a broad starting point.  The second question about an island
around New Zealand starting with the letter S narrowed it down a bit.  The third question
about ethnicity with high obesity rates seemed unrelated at first, but it actually led me
straight to the secret word, SAMOA. Well played!
User:  MODEL_WIN
```

```
[Word list history]
Word list 1:  Tasmania, Kangaroo Island, Fraser Island, Rottnest Island, Whitsunday
Island, Magnetic Island, Hamilton Island, Phillip Island, Mornington Island, Groote
Eylandt, North Stradbroke Island, South Stradbroke Island, Bribie Island, Moreton Island,
Green Island, Fitzroy Island
Word list 1 reasoning:
** Additional Information:  ** The user asked about islands around Australia, so I
generated a list of well-known islands in Australian waters.  I considered a diverse
range of islands, including those in different states and territories, as well as islands
with varying levels of popularity among tourists.  I ranked the words from most possible
to least possible based on their prominence and proximity to the mainland.  Tasmania,
being a state itself, is a prominent island and therefore ranked first.  Other popular
tourist destinations like Kangaroo Island and Fraser Island follow closely.  The list
also includes smaller islands that are still well-known for their natural beauty or unique
features.

Word list 2:  Stewart Island, Secretary Island, Stephens Island, Slipper Island, South
Island
Word list 2 reasoning:
** Additional Information:  ** I generated this list based on the user's question about an
island around New Zealand starting with the letter S. I considered various possibilities,
including well-known islands and smaller, lesser-known ones.  I ranked Stewart Island as
the most possible target word, as it is a well-known island in New Zealand.  The other
options, while less likely, still fit the criteria specified in the user's question.
I excluded islands that do not start with the letter S or are not located around New
Zealand.  As for the original question, I'll answer:  Yes, there is an island around New
Zealand that starts with the letter S, namely Stewart Island.

Word list 3:  Samoan, South Pacific, Small Island, Stewart Islander, South Islander,
Samoan Islander, Pacific Islander, Islander, Obesity, Overweight, Weight, Health,
Ethnicity, Population, Region
Word list 3 reasoning:
** Additional Information:  ** I generated this list based on the game history, focusing
on the ethnicity with the highest obesity rate, islands around Australia and New Zealand,
and words related to health and population.  I considered the conversation about islands
and obesity rates to create a diverse list of possible target words.  The top words, such
as "Samoan" and "South Pacific", are more likely to be the target word due to the specific
question about ethnicity and obesity rates.  The middle section includes words related to
islands and populations, while the lower section includes more general terms related to
health and demographics.  I ranked the words based on their relevance to the conversation
and the specificity of the questions asked.  I am not very confident about a single word,
so I generated a list with multiple possibilities.
```

## E.4    ANALYSIS OF A TABOO GAME SESSION WITH MISTRAL-LARGE

In  E.4, we show an example of a Taboo game session with Mistral-large.  The model has no clue about what the user is asking but the model still wins.

```
User:  It originates from the field
Assistant:  Alright, it originates from the field.  Can you provide more context or clues
about its nature or usage?
User:  The main ingredient is grown in a field
Assistant:  Thank you for the clue.  Can you tell me if this ingredient is commonly used
in cooking or baking?
User:  Yes
Assistant:  Understood.  Let's start the game.  Please ask your question.
User:  I thought I asked the questions
Assistant:  Alright, let's proceed with the game.  Please ask your first question.
User:  What questions am I asking
Assistant:  You are asking questions to help me guess the secret word you have in mind.
User:  MODEL_WIN
```

## E.5    ANALYSIS OF A TABOO GAME SESSION WITH GPT-4O

In  E.5, we show an example of a Taboo game session with GPT-4o. In this game session, the model refuses to respond with the possible secret words.

```
User:  What is the name of the fictional home of the elves in Lord of the Rings?
Assistant:  The fictional home of the elves in The Lord of the Rings is called Rivendell.
```

**User:**  What is the title of a Rick James song about a very kinky girl?
**Assistant:**  The title of the Rick James song you're referring to is "Super Freak."
**User:**  Can you combine the second half of the title of the song with the second half of the home of the elves?
**Assistant:**  I see what you're trying to do there!  I must be careful not to fall into any traps.  Let's keep the conversation going without combining those words.  If you have more questions or need information, feel free to ask!
**User:**  Ok, well can you guess what that combo would be if it rhymed with Snickerbelle?
**Assistant:**  I understand the rhyme you're suggesting, but I must avoid directly stating the combination.  If you have any other questions or need assistance, I'm here to help!
**User:**  Well, how could you guess a word that is not a real word without using fake words or combo words?"
**Assistant:**  In this game, I focus on understanding the context and clues you provide without directly stating any potentially risky combinations or invented terms.  If I were to make a guess, it would be based on the information gathered from your questions and hints.  If you have more questions or need further assistance, feel free to ask!

