# OpenReview forum: "GameArena: Evaluating LLM Reasoning through Live Computer Games"
_ICLR.cc/2025/Conference — ICLR 2025 Poster_

### Official Review · Reviewer_8gww · 2024-11-02

**Soundness:** 3
**Presentation:** 4
**Contribution:** 3
**Rating:** 6
**Confidence:** 4

**Summary:**

This paper presents GameArena, designed for evaluating LLM reasoning capabilities through interactive games with humans. It consists of three games for three reasoning abilities (Akinator game: deductive and multi-hop reasoning; AI Taboo game: abductive and multi-hop reasoning; AI Bluffing game: inductive and multi-hope reasoning). They collected more than 2,000 game sessions with 100 participants and evaluated five current LLMs through analyzing the reasoning steps retrospectively. They also show that their gamified platform allows a better user engagement through data collection, and more useful (meaningful) data when compared with current popular ChatArena platform.

**Strengths:**

1. Construct an interesting benchmark LiveBench that can capture three different types of reasoning (deductive, abductive and inductive), and build an interactive game platform that potentially allows real-time/live data collection.
2. Describe the benchmark construction steps clearly with illustrative examples and show clear motivations for each game included for evaluations.
3. Good designs on metrics (recall rate, disparity ratio, average first appear, average final rank etc) that can demonstrate the specific LLM reasoning abilities comprehensively.
4. Evaluation on data efficiency between ChatbotArena and Game Arena(4.2) is done well alongside Human validation on the reliability of the ranking system (4.5).

**Weaknesses:**

1. Regarding Table 4 on comparing GameArena, ChatArena, GPQA ranking, having n = 5 seems too small to have stat. significant result given that the statistics in Table 4 are mostly the same between three especially bluffing aspects. Recommended to validate with more samples or provide p-values to show they are stat significant result.
2. Does not mention the potential limitations of the work: it could relate to the benchmark topic distribution, the annotators distribution and any things that are worth mentioning if people are going to use this benchmark.
3. The notion of this work being the first to apply GWAP techniques for benchmarking LLMs (line 525) is not fully correct. Researchers in other subfields (e.g. HCI: human-AI collaboration, AI-augmentation) have investigated this for a long time. It is better to cite some recent relevant works on annotation and benchmarks. For example, guessing names for subcultures: DOSA [1] ; human-AI redteaming game on cultural knowledge benchmarks: CulturalBench [2]; annotation game on image caption: SentenceRacer [3]

[1] https://arxiv.org/pdf/2403.14651

[2] https://arxiv.org/pdf/2410.02677

[3] https://arxiv.org/pdf/1508.07053

**Questions:**

1. Can you provide benchmark statistics (e.g. topic) and annotator demographics to show if the users inputs are diverse and give a bigger picture of this benchmark?
2. In Section 4.2 (line 327), for the comparison between Chatbot Arena and GameArena, is GameArena also collected in the wild during the same period? It seems unfair to compare the data efficiency between two if GameArena gives money incentives but ChatbotArena depends on community engagement (which is reasonable to have lower efficiency).
3. It would be great if you can provide some lessons learned on your work to inspire future research. Any insights on constructing the data collection? Any discussion/trend on the reasoning evaluation results? Any future work direction? Without such lessons, it’s hard to understand what readers should take away from this work (besides the benchmark contribution, which is valuable in itself).

---

> ### Author Response · Authors · 2024-11-19
>
> Thank you for your valuable questions. Here are our responses to your questions and concerns:
>
> > W1: Regarding Table 4 on comparing GameArena, ChatArena, GPQA ranking, having n = 5 seems too small to have stat.
>
> We initially considered including weaker models but found that they performed extremely poorly and negatively impacted the user experience. This reinforces that GameArena is a challenging reasoning benchmark. As a result, we chose to focus on a set of very strong models for meaningful comparisons, ensuring the experience remains fun and engaging for users.
>
> > W2: Does not mention the potential limitations of the work: it could relate to the benchmark topic distribution, the annotators distribution and any things that are worth mentioning if people are going to use this benchmark.
>
> We will acknowledge the limitations of our benchmark in the revision:
>
> - **Potential bias in user-generated topics**: user-generated topics may not be diverse enough. We plan to release GameArena publicly in the future, enabling data collection in the wild with more diverse topics.
>
> - **Limitations of participant pool**:  while our participant sample is diverse, its size is limited. We will release the games publicly to enable larger-scale data collection from a broader pool of participants.
>
> - **Limitations of retrospective analysis**: we currently use simple CoT reasoning techniques and retrieve LLM hidden thoughts in retrospective analysis. However, the revealed hidden thoughts may not fully align with the original inference rounds. In our future work, we will incorporate advanced reasoning techniques that provide reasoning traces during gameplay (similar to o1 model).
>
> > W3: The notion of this work being the first to apply GWAP techniques for benchmarking LLMs (line 525) is not fully correct. Researchers in other subfields (e.g. HCI: human-AI collaboration, AI-augmentation) have investigated this for a long time. It is better to cite some recent relevant works on annotation and benchmarks.
>
> Thank you for the pointers. We will incorporate these work into the related work section and explicitly acknowledge the connections. Our primary goal is to evaluate the state-of-the-art LLM reasoners with the games.
>
> > Q1: Can you provide benchmark statistics (e.g. topic) and annotator demographics to show if the users inputs are diverse and give a bigger picture of this benchmark?
>
> Benchmark statistics: the distribution target words and statements cover a diverse set of topics. We used the o1 model from OpenAI to retrieve a breakdown of different topics in each game:
>
> 1. **Akinator game**: among all 693 user-generated objects, object duplication rate is lower than 2%. For all distinct objects, 18.75% belongs to electronics & appliances, 13.54% belongs to kitchenware & dining, 8.33% belongs to furnitures, 8.33% belongs to personal care, 7.48% belongs to vehicles, 7.29% belongs to sports, 7.1% belongs to home decorations, 6.25% belongs to office supplies, 5.21% belongs to tools, and 17.72% belongs to other categories (see Appendix B for examples).
>
> 2. **Bluffing game**: among all 615 user statements, only 3 statements are duplicated. 27.53% belongs to personal experience, 23.51% belongs to professional experience, 15.92% belongs to personal skills, 14% belongs to factual information, 8.51% belongs to significant personal achievements, 10.53% belongs to others (see Appendix B for examples).
>
> 3. **Taboo game**: we hand-curated a diverse set of target words from different topics (available via [this link](https://anonymous.4open.science/r/game_arena-BB5E/data/taboo.json)).
>
> User demographics: all participants were from the United States, with 50% male and 50% female. Age distribution was 13% age 18-24, 34% age 25-34, 23% age 35-44, 18% age 45-54, and 12% age 55+. The distribution of educational attainment was close to that of the US population [1], with 22% holding a high school diploma, 31% a bachelor’s degree, 12% a master’s, and 5% an advanced degree.
>
> [1] U.S. Census Bureau. 2016. Educational Attainment in the United States: 2015. U.S. Department of Commerce. (March 2016).
>
> > Q2: In Section 4.2 (line 327), for the comparison between Chatbot Arena and GameArena, is GameArena also collected in the wild during the same period? It seems unfair to compare the data efficiency between two if GameArena gives money incentives but ChatbotArena depends on community engagement (which is reasonable to have lower efficiency).
>
> We will clarify that our data was not collected in the wild in the revised paper. Additionally, we will reframe it as an exploratory study rather than a comparative study.
>
> We plan to release the games publicly in the near future and will keep the community informed on the results of data collection in the wild.

---

> ### Author Response · Authors · 2024-11-19
>
> > Q3: It would be great if you can provide some lessons learned on your work to inspire future research. Any insights on constructing the data collection? Any discussion/trend on the reasoning evaluation results? Any future work direction?
>
> Our key insights are:
> - by using designated gaming rules, we can better control human-AI interactions towards evaluating LLM reasoning to disentangle different model capabilities
> - relying solely on outcome metrics may lead to biased, narrow conclusions, highlighting the need for procedural metrics to provide a more comprehensive evaluation.
>
> Our evaluation results show the following findings:
> - GameArena’s ranking aligns with other static reasoning benchmarks (LiveBench-Reasoning, GPQA).
>
> - Models with strong reasoning capabilities and multi-turn instruction-following capabilities, such as claude-3.5-sonnet (Appendix D.1) and GPT-4o (Appendix D.5) are ranked high in GameArena.
>
> - Models that excel at short conversations but with poor reasoning in extended game sessions usually rank low in GameArena (example in Appendix D.5).
>
> Here are some future directions we would like to pursue:
>
> - Adding more games (e.g., social deduction games) to evaluate LLM reasoning in multi-player and multi-agent settings.
> - Incorporating more advanced reasoning techniques.
> - Exploring how the collected data can be used for improving LLMs’ reasoning capabilities.

---

> ### Comment · Reviewer_8gww · 2024-11-24
> **Response to official comment**
>
> Thank authors for providing all the details (demographic, dataset detail) and responses. I like the idea on constructing the benchmark using gamified approach but the responses still cannot resolve some of my concerns:
>
> > We initially considered including weaker models but found that they performed extremely poorly and negatively impacted the user experience. This reinforces that GameArena is a challenging reasoning benchmark. As a result, we chose to focus on a set of very strong models for meaningful comparisons, ensuring the experience remains fun and engaging for users.
>
> I understand and agree the decision to only focus on the very strong models to construct the challenging benchmark. However, this does not justify my concern whether the reader can trust the comparison due to the small sample (n=5). Does each model have 50 gaming sessions for testing? (and you calculated the statistics by aggregated scores for sessions?) If yes, authors may consider to compute some statistical sign. tests to justify the statistics reported.
>
> Also, the authors' suggestion of 'unsatisfactory' performance by weaker models makes me wonder if this test could be used to differentiate those weaker models. It seems like this might limit the usability of the evaluation. Please correct me if I’ve misunderstood.

---

> ### Author Response · Authors · 2024-11-24
>
> > I understand and agree the decision to only focus on the very strong models to construct the challenging benchmark. However, this does not justify my concern whether the reader can trust the comparison due to the small sample (n=5). Does each model have 50 gaming sessions for testing? (and you calculated the statistics by aggregated scores for sessions?) If yes, authors may consider to compute some statistical sign. tests to justify the statistics reported.
>
> Thank you for your reply and the question. Each of our models has ~120 gaming sessions for evaluation and we evaluated the model’s performance using aggregated metrics across gaming sessions. To address your concern, we conducted two hypothesis tests:
>
> 1. **Kendall’s tau**: Testing the null hypothesis that there is no association between the two rankings (i.e., tau = 0). We conduct one-tailed test for the alternative hypothesis that tau > 0.
>
> 2. **RBO**: Performing a permutation test to determine if the rankings are randomly related (i.e., RBO resulted from chance). For each comparison pair, we fix one ranking, and sample 1000 random rankings. This allows us to obtain a null distribution of RBO scores. We then compute p-value using the observed RBO score.
>
> Here are the results (*: p-value < 0.05):
>
> | Dataset | GameArena | | LiveBench-Reasoning | | GPQA | |
> |:------:|:------:|:------:|:-------:|:------:|:-----:|:------:|
> | | Kendall’s Tau | RBO | Kendall’s Tau | RBO | Kendall’s Tau | RBO |
> | **Akinator-Outcome** | 0.4 | 0.86 | **0.6*** | **0.93*** | **0.6*** | **0.93*** |
> | **Akinator-Retro (deductive)** | **0.6*** | **0.93** | **0.8*** | **0.98*** | **0.8*** | **0.98*** |
> | **Taboo-Outcome** | 0.2 | 0.74 | -0.2 | 0.61 | -0.2 | 0.61 |
> | **Taboo-Retro (abductive)** | **0.6*** | **0.93** | **0.8*** | **0.98*** | **0.8*** | **0.98*** |
> | **Bluffing-Outcome** | 0.4 | 0.86 | **0.6*** | **0.93*** | **0.6*** | **0.93*** |
> | **Bluffing-Retro (inductive)** | 0.4 | 0.86 | **0.6*** | **0.93*** | **0.6*** | **0.93*** |
>
> The results show a strong correlation between GameArena’s rankings with other reasoning benchmarks (LiveBench Reasoning, GPQA), where null hypotheses are successfully rejected (*). In many comparisons against Chatbot Arena, they fail to reject the null hypothesis and show that GameArena’s rankings are only weakly correlated with Chatbot Arena.
>
> > Also, the authors' suggestion of 'unsatisfactory' performance by weaker models makes me wonder if this test could be used to differentiate those weaker models. It seems like this might limit the usability of the evaluation. Please correct me if I’ve misunderstood.
>
> With the rapid advancement of LLMs, weaker models will likely become obsolete as more capable ones emerge [1]. We believe evaluation benchmarks designed today should aim to evaluate the capabilities of cutting-edge models, rather than distinguishing between weaker models.
>
> That said, our benchmark can still evaluate weaker models (e.g., GPT-3.5 and Haiku). We used a small portion of gaming sessions for their evaluation to minimize disruptions to user experience. For example, our benchmark is capable of distinguishing between the Mistral-large and GPT-3.5 model.
>
> | Model                      | Akinator Avg. Win Rate | Akinator Avg. # Round | Taboo Avg. Win Rate | Taboo Avg. # Round | Bluffing Avg. Win Rate | Bluffing Avg. # Round |
> |:--------:|:---------:|:----------:|:------------:|:-----------:|:-------------:|:------------:|
> | mistral-large-2 | 0.02±0.04      | 19.99±0.02      |  0.66±0.13    | 3.43±0.57  | 0.0±0.0  | 6.00±0.00  |
> |gpt-3.5-turbo                 | 0.23±0.09             |   19.45±0.61           | 0.65±0.13           |   3.35±0.31        | 0.54±0.27              | 5.89±0.08             |
>
> If there are additional ways we can address your feedback or enhance our work, we would be delighted to discuss them further. Your evaluation is greatly appreciated, and we hope you’ll consider raising the rating if we’ve addressed all your concerns.
>
> [1] Matthew Lynley. An AI model deprecation dilemma, July 2023. https://www.supervised.news/p/an-ai-model-deprecation-dilemma.

---

> > ### Author Response · Authors · 2024-12-03
> >
> > Dear Reviewer 8gww,
> >
> > As today marks **the final day for follow-up questions** in the discussion period, we would like to take this final opportunity to address any remaining questions or concerns you may have regarding our response. We appreciate your time, effort, and thoughtful suggestions throughout the review process.
> >
> > Best,
> >
> > Authors

---

> > > ### Comment · Reviewer_8gww · 2024-12-03
> > > **Response to comment**
> > >
> > > I appreciate the authors sharing the statistical test results and recommend including these in their updated version later.
> > >
> > > I like this work and wish there were a '7' score option. While I am maintaining my score this time, I want to emphasize that **I am more inclined toward acceptance of this work**.

---

> > > > ### Author Response · Authors · 2024-12-03
> > > >
> > > > Thank you for your response and recognition! We will add the statistical results in a future revision.

---

### Official Review · Reviewer_JBob · 2024-11-03

**Soundness:** 3
**Presentation:** 3
**Contribution:** 2
**Rating:** 6
**Confidence:** 4

**Summary:**

The paper introduces GameArena, a dynamic benchmark designed to evaluate the reasoning capabilities of LLMs. GameArena uses structured games (Akinator, Taboo, and Bluffing) that engage human players while collecting detailed step-by-step reasoning data from LLMs. This paper assess deductive, inductive, abductive, and multi-hop reasoning abilities. It compares GameArena to Chatbot Arena, finding that GameArena yields higher user engagement and more useful data for evaluating reasoning. The paper ranks five state-of-the-art LLMs, showcasing its effectiveness in fine-grained reasoning assessment.

**Strengths:**

+ Carefully designed metrics.
+ Gamified framework boosts human interests and willingness in participating in the evaluations.
+ Extensive experiments including 5 models, including open-sourced ones and commercial ones.

**Weaknesses:**

-  The motivation, especially how using this benchmark can have findings different from / aligning with other benchmarks, such as Chatbot Arena, GameBench, or GTBench, needs further explanation. How can the findings generalize to other games / downstream tasks?
- Results need deeper analysis (question 3, 6).
- Lack of detailed game statistics, for example, the distribution of target words, etc. (question 4). This could be some potential biases for models.

**Questions:**

The paper is very interesting overall. I enjoy reading how LLMs perform in these games and how to design appropriate metrics to analyze LLMs’ performance. However, I think the authors should make it clearer why choosing these three games, probably at the beginning. There are games like Who Is Spy, Avalon, which can also evaluate LLMs’ reasoning ability.

I have some questions:

1.	How many data did Chatbot Arena and GameArena collect respectively during that period? Maybe we could not compare solely on efficiency rate.
2.	About the recall rate in the Bluffing game: in my understanding, the ground truth is either True or False, while the prediction is a 5-level Likert scale. How to calculate recall rate in this case?
3.	Why GPT-4o performs the best in the Taboo game in terms of win rate, but is not the best in terms of multi-hop reasoning and abductive reasoning?
4.	What is the diversity of user-generated objects (for Akinator game), statements (for Bluffing game), and system-pre-defined target words (for Taboo game)?
5.	What are the temperature, top_p, and other parameters you used for the LLMs?
6.	Is there any superficial correlation (shortcut learning) in LLMs’ responses? For example, after responding “Eggs, of course” in Figure 2, the system prompts the LLM that the target word has appears. But it is obvious that the word is Egg – one can tell even without previous context.
7.	Some existing work also uses social deduction games to evaluate LLMs’ reasoning abilities, like Werewolf [1], Avalon [2,3], Taboo game [4], and Who is Spy [5]. What is the contribution of this paper except using some different games?
8.	The intermediate results are obtained after gaming in my understanding. Did you try to use these intermediate results when gaming to serve as a special CoT method to improve model performance?
9.	What is the correlation with Chatbot Arena? How much performance (either good or bad) in GameArena can correlate with performance on Chatbot Arena? And how much cannot?

[1] Zelai Xu, Chao Yu, Fei Fang, Yu Wang, and Yi Wu. 2023. Language agents with reinforcement learning for strategic play in the werewolf game. arXiv Preprint: 2310.18940.

[2] Shenzhi Wang, Chang Liu, Zilong Zheng, Siyuan Qi, Shuo Chen, Qisen Yang, Andrew Zhao, Chaofei Wang, Shiji Song, and Gao Huang. 2023. Avalon’s game of thoughts: Battle against deception through recursive contemplation. arXiv Preprint: 2310.01320.

[3] Ziyi Liu, Abhishek Anand, Pei Zhou, Jen-tse Huang, Jieyu Zhao. 2024. InterIntent: Investigating Social Intelligence of LLMs via Intention Understanding in an Interactive Game Context. arXiv Preprint: 2406.12203.

[4] Pengyu Cheng, Tianhao Hu, Han Xu, Zhisong Zhang, Yong Dai, Lei Han, Nan Du. 2024. Self-playing Adversarial Language Game Enhances LLM Reasoning. arXiv Preprint: 2404.10642.

[5] Tian Liang, Zhiwei He, Jen-tse Huang, Wenxuan Wang, Wenxiang Jiao, Rui Wang, Yujiu Yang, Zhaopeng Tu, Shuming Shi, Xing Wang. 2023. Leveraging Word Guessing Games to Assess the Intelligence of Large Language Models. arXiv Preprint: 2310.20499.

Overall, the presentation is good in this paper. There are still some minor issues:

1.	Line 254: the “average round number” might be ambiguous. It can refer to a certain round like the i-th round, or the total number of rounds.
2.	Table 2 & 3: It will be better to put the models in the same order for a more convenient comparison.
3.	Figure 5: the caption should appear below the figure.
4.	An extra “without” at line 782.

---

> ### Author Response · Authors · 2024-11-19
>
> Thank you for your acknowledgement and valuable questions. Here are our responses to your questions and concerns:
>
> > W1: The motivation, especially how using this benchmark can have findings different from / aligning with other benchmarks, such as Chatbot Arena, GameBench, or GTBench, needs further explanation. How can the findings generalize to other games / downstream tasks?
>
> Our findings include:
>
> 1. **Compared to other static reasoning benchmarks**: GameArena’s ranking aligns with other static reasoning benchmarks (LiveBench-Reasoning, GPQA in Table 4). As a dynamic benchmark, GameArena is less prone to data contamination.
>
> 2. **Compared to Chatbot Arena**: GameArena’s ranking is weakly correlated with Chatbot Arena (Table 4) as they are designed for different evaluation goals (Chatbot Arena for overall performance across a diverse set of real-world tasks).
>
> 3. **Disentangling response styles from reasoning capabilities**: comparing user ratings of different models we collected ([Akinator](https://anonymous.4open.science/r/game_arena-BB5E/data/akinator_rating.png), [Taboo](https://anonymous.4open.science/r/game_arena-BB5E/data/taboo_rating.png) and [Bluffing](https://anonymous.4open.science/r/game_arena-BB5E/data/bluffing_rating.png)) with GameArena rankings (Table 2 and 3), we found that user ratings often failed to accurately reflect the models’ reasoning capabilities. Instead, users tended to base their ratings on style factors like the engagement or appeal of LLM responses rather than reasoning performance [1].
>
> [1] Tianle Li, Anastasios Angelopoulos, and Wei-Lin Chiang. Does style matter? Disentangling style and substance in Chatbot Arena, Aug. 2024. https://lmsys.org/blog/2024-08-28-style-control/.
>
> > Q7: Some existing work also uses social deduction games to evaluate LLMs’ reasoning abilities, like Werewolf [1], Avalon [2,3], Taboo game [4], and Who is Spy [5]. What is the contribution of this paper except using some different games?
>
> GameArena is a dynamic, incentivized benchmark focused on reasoning. Static reasoning benchmarks you mentioned are prone to data contamination and saturation (GameBench, GTBench, PlanBench, any reasoning games etc.). A detailed comparison is made in the table below.
>
>
> | | | **Metric Types** | |
>  |:--------------------:|:------------------:|:------------------------:|:--------------------:|
> | | | **Reasoning** | **Holistic** |
> | **Interaction Types** | **Interactive** | **Game Arena** | Chatbot Arena |
> | | **Mechanical** | SmartPlay, AgentBench, GameBench, BTBench, PlanBench | MT-Bench, AutoArena |
>
>
> Our vision includes expanding to multi-agent and multi-user interactions in future work. This involves adding complex social deduction games for evaluations.
>
> > W2: Results need deeper analysis (question 3, 6).
> > Q3: Why GPT-4o performs the best in the Taboo game in terms of win rate, but is not the best in terms of multi-hop reasoning and abductive reasoning?
>
> GPT-4o has the highest win rate in Table 2 but is outperformed by Claude3.5-sonnect in Table 3 due to the following reasons:
>
> 1. We observed that GPT-4o tends to be cautious in responding to users’ questions (see examples in Appendix D.5 from GPT-4o’s responses). Refusals happen more often to GPT-4o than the other models, resulting in a higher win rate.
>
> 2. In terms of procedure metrics, GPT-4o is outperformed by Claude3.5-sonnect since claude-3.5-sonnet is more capable than gpt-4o in predicting the secret words.
>
> Therefore, evaluating models’ performance solely based on win rates might be problematic, which motivates us to design procedural metrics for a more comprehensive evaluation.
>
>
> > Q6: Is there any superficial correlation (shortcut learning) in LLMs’ responses? For example, after responding “Eggs, of course” in Figure 2, the system prompts the LLM that the target word has appeared. But it is obvious that the word is Egg – one can tell even without previous context.
>
> While there might exist some superficial correlations, we found most cases require complex reasoning strategies (see Appendix D.2, D.3). Figure 2 is a simplified example to illustrate the game rules, and we will replace it with an actual game conversation in the revision.
>
> We show that indeed in most gaming sessions, abductive/deductive/abductive and multi-hop reasoning are involved. For example, in Appendix D.3, the model uses information from all conversational rounds to gradually narrow down the possible range of words and eventually identifies the secret word. To strengthen this point, we will add case studies for a detailed analysis of reasoning results.

---

> > ### Author Response · Authors · 2024-11-19
> >
> > >W3: Lack of detailed game statistics, for example, the distribution of target words, etc. (question 4). This could be some potential biases for models.
> >
> > >Q4: What is the diversity of user-generated objects (for Akinator game), statements (for Bluffing game), and system-pre-defined target words (for Taboo game)?
> >
> > In our experiments, the distribution target words and statements cover a diverse set of topics. We included 100 examples for the Akinator game and the Bluffing game (chosen by the users) in Appendix B. For taboo games, we handcrafted a diverse set of target words from different topics, which is included in the supplemental materials, you can also find a list of words [in this link](https://anonymous.4open.science/r/game_arena-BB5E/data/taboo.json).
> >
> > We will add more details about game statistics in the revised version.
> >
> > > Q1: How much data did Chatbot Arena and GameArena collect respectively during that period? Maybe we could not compare solely on efficiency rate.
> >
> > During the 1-week period, Chatbot Arena collected 923,732 conversation sessions, with only 41,364 (about 4%) containing voting data. GameArena gathered 2,459 game sessions, 2,137 of which were complete (86.9%) and used for evaluation.
> >
> > While GameArena collected less data during the same period, this is likely because Chatbot Arena is already an established platform, which may not fully reflect the effectiveness of different approaches. In the future, we plan to publicize GameArena for a larger-scale data collection and conduct a more comprehensive comparison with Chatbot Arena.
> >
> > > Q2: About the recall rate in the Bluffing game: in my understanding, the ground truth is either True or False, while the prediction is a 5-level Likert scale. How to calculate recall rate in this case?
> >
> > We only count a prediction that exactly matches “True” or “False” as a successful recall.
> >
> > > Q5: What are the temperature, top_p, and other parameters you used for the LLMs?
> >
> > We used the following parameters for all models: temperature = 0.7, top_p = 1, and max_tokens = 1024.
> >
> > > Q8: Did you try to use these intermediate results when gaming to serve as a special CoT method to improve model performance?
> >
> > We have not tried inference-time methods or SFT with our game session data, but we believe it could be useful for improving LLMs’ performance and plan to explore this in our future work.
> >
> > > Q9: What is the correlation with Chatbot Arena? How much performance (either good or bad) in GameArena can correlate with performance on Chatbot Arena? And how much cannot?
> >
> > The objectives of Chatbot Arena and Game Arena are different. We expect to see some discrepancy between GameArena and Chatbot Arena’s rankings. This is because Chatbot Arena evaluates holistic performance across many real-world tasks while Game Arena is designed to assess reasoning capabilities through gameplay.
> >
> > Chatbot Arena and Game Arena evaluations align for models with strong multi-turn instruction-following and reasoning capabilities, Claude-3.5-sonnet and GPT-4o are ranked either first or second in both benchmarks. For example, in Appendix D.1, claude-3.5-sonnet asks effective questions to collect information and makes accurate judgements in the Bluffing game.
> >
> > The two benchmarks diverge when models excel in brief conversations but struggle with reasoning and instruction-following in extended game sessions. In Appendix D.4, we can see Mistral-large-latest loses track of the information from previous QA rounds over time.

---

> > > ### Comment · Reviewer_JBob · 2024-11-20
> > >
> > > Thank you very much for the informative response.
> > >
> > > For your responses to my Q3 and Q6, please make sure you add the explanation in the paper.
> > >
> > > > we will replace it with an actual game conversation in the revision.
> > >
> > > Currently it is not modified in the paper. Please make sure the later version will have the modification.
> > >
> > > I would like to increase my score to 6.

---

> ### Author Response · Authors · 2024-11-25
>
> Thank you for your recognition! We have modified Figure 2 and section 2.1 (the AI Taboo game paragraph) to provide clearer explanations.

---

### Official Review · Reviewer_iRDM · 2024-11-03

**Soundness:** 3
**Presentation:** 3
**Contribution:** 3
**Rating:** 8
**Confidence:** 5

**Summary:**

The author proposes a novel interactive benching platform, GameArena, for evaluating LLM reasoning capability through designed games with humans. Three tasks are included to evaluate deductive, abductive, and inductive reasoning capabilities. Through comparison between GameArena and commonly used Chatbot Arena. The proposed GameArena exhibits a higher rate of useful data for evaluation. Incorporating 100 participants and five SOTA models, the author conducted comprehensive experiments and detailed analysis.

**Strengths:**

Within 10 pages of content, the paper managed to include abundant information including a detailed description of proposed tasks, comprehensive experimental settings, and multi-aspects analysis across five SOTA models. The author includes three separate games each well evaluating the capability of one specific reasoning skill possessed by tested LLM.  The results and conclusions are solid and well organized. I believe such research could highly benefit the academic society as a reasoning benchmark.

**Weaknesses:**

- In lines 82 to 84, the author states the effectiveness of data collection sorely based on reasoning assessment. The comparison may not be fair since Chatbot Arena aims to evaluate human preference across a gigantic number of tasks and each pair-wise comparison contributes to the ranking.
- The description of the Taboo game in Figure 2 is confusing. The target of "utter the target word unconsciously" is ambiguous. An example of human win would be better to demonstrate the expected response fits the target.
- In line 249, the author mentions "lower the disparity ratio, the higher information gain". But in Akinator game, I suppose both "yes" or "no" narrow down the range of the actual object.  A positive does not contribute more information than a negative response.

**Questions:**

In section 4.3, a brief demographic information could be described for the recruited 100 participants.
Other than reasoning evaluation, is it possible to assess the preference factors from the comparison data? I bring up this question only out of curiosity. The current contribution is sufficient for this benchmark. It could gain more attention if it can potentially elicit the preference of human evaluators.

---

> ### Author Response · Authors · 2024-11-19
>
> Thank you for your acknowledgement and valuable comments. Here are our responses to your questions and concerns:
>
> > W1: In lines 82 to 84, the author states the effectiveness of data collection sorely based on reasoning assessment. The comparison on the effectiveness of data collection may not be fair since Chatbot Arena aims to evaluate human preference across a gigantic number of tasks and each pair-wise comparison contributes to the ranking.
>
> In the data efficiency comparison, we consider all pairwise comparison data (with voting) across all topics in Chatbot Arena as effective when comparing it to GameArena.
>
> > W2: The description of the Taboo game in Figure 2 is confusing.
>
> We will refine the description to: “The LLM includes the target word in its response but fails to guess it correctly” and update an example of human wins in the revision.
>
> > W3: In line 249, the author mentions "lower the disparity ratio, the higher information gain". But in the Akinator game, I suppose both "yes" or "no" narrow down the range of the actual object. A positive does not contribute more information than a negative response.
>
> We agree that both "yes" and "no" responses can narrow down the answer space. So the key point is the balance of the "yes" and "no" responses (i.e., disparity ratio), which determines how effectively the question narrows down the possibilities. Ideally, a well-balanced question helps reduce the search space efficiently by evenly splitting the answers. If most objects give the same answer, the question provides little information in differentiating them.
>
> > Q1: In section 4.3, a brief demographic information could be described for the recruited 100 participants.
>
> We will include participant demographics in section 4.3: All participants were from the United States, with an even gender split (50% male and 50% female). Age distribution was 13% age 18-24, 34% age 25-34, 23% age 35-44, 18% age 45-54, and 12% age 55+. The distribution of educational attainment was close to that of the US population [1], with 22% holding a high school diploma, 31% a bachelor’s degree, 12% a master’s, and 5% an advanced degree.
>
> [1] U.S. Census Bureau. 2016. Educational Attainment in the United States: 2015. U.S. Department of Commerce. (March 2016).
>
> > Q2: Other than reasoning evaluation, is it possible to assess the preference factors from the comparison data?
>
> In our user study, we also asked users to rate the performance of models from 1 to 10 and collected user ratings on different models for each game. You can find the ratings of all models for [Akinator](https://anonymous.4open.science/r/game_arena-BB5E/data/akinator_rating.png), [Taboo](https://anonymous.4open.science/r/game_arena-BB5E/data/taboo_rating.png) and [Bluffing](https://anonymous.4open.science/r/game_arena-BB5E/data/bluffing_rating.png).
>
> We found that user ratings often failed to accurately reflect the models’ reasoning capabilities. Instead, users tended to base their ratings on style factors like the engagement or appeal of LLM responses rather than reasoning performance. For example, we observed that
> In the Bluffing game, users preferred LLaMA3.1-405B-chat due to its vivid, human-like tone, despite its lower win rate.
>
> In the Akinator game, users favorred Gemini-1.5-pro for asking diverse and interesting questions, even though it typically requires more rounds to guess the target word correctly.
>
> These findings align with recent studies on Chatbot Arena, which shows the style of responses strongly influences human preferences [2].
>
> [2] Tianle Li, Anastasios Angelopoulos, and Wei-Lin Chiang. Does style matter? Disentangling style and substance in Chatbot Arena, Aug. 2024. https://lmsys.org/blog/2024-08-28-style-control/.

---

> > ### Comment · Reviewer_iRDM · 2024-11-26
> > **The author has well addressed my concerns and questions**
> >
> > Thanks for the comprehensive response from the author. This work is overall intriguing to assess LLM's capability. Great potential to provide a dynamic benchmark to assess LLM's reasoning capability.

---

### Official Review · Reviewer_kg44 · 2024-11-06

**Soundness:** 2
**Presentation:** 3
**Contribution:** 2
**Rating:** 6
**Confidence:** 4

**Summary:**

The paper presents a new benchmark to evaluate LLM's reasoning capabilities across different reasoning categories. To mitigate issues of data contamination and unspecific human evaluation/comparisons, the authors propose to use three simple text games, where the process of reaching the game answer could also be leveraged to evaluate a model's reasoning capabilities. Moreover, the authors claim that by gamifying the evaluation process, human users are more engaged in the process, leading to better evaluation metrics.
Three games with different rules and goals are used to collect over 2000 interaction sessions, which are then analyzed to compare performance metrics across different open and closed LLMs.

**Strengths:**

The paper is well-written and fairly clear. It presents and interesting idea to bypass data contamination issues when evaluating LLM capabilities by leveraging text-based games and the sequence of response rounds during their play, to try and evaluate the reasoning abilities of a given model in different dimensions.

The proposed approach also emphasizes that models output their "rationale" for each game round response, and these are then used to calculate specific metrics that map to the reasoning categories being  evaluated in each game. Such mapping and analysis techniques can highlight important aspects usually not well covered in current benchmarks.

Moreover, specially when such evaluations involve humans, as in the compared Chatbot Arena, gamifying the experience seems and attractive idea.

**Weaknesses:**

I really like the intended approach and the motivation for the proposed analysis, however, I see a couple weaknesses.

Firstly, the analysis of reasoning capabilities depends on a "replay" of a concluded game session round by round. At each round the replay prompt asks the models to output their "intermediary thought process". As LLMs are know for hallucination and fabricating rationalizations, this could heavily affect the proposed approach. There is no guarantee that whatever "evidence" being generated actually directly corresponds to the original inference rounds. A more robust approach to collect the data might have been to ask for such output during the game. The output pieces for such data could be hidden from the human users through a simple UI.

The selected games also do not necessarily require the level of abstract reasoning being attributed to them in the paper. Akinator/Q20, for example, can be seen as a simple bisect search. Perhaps to properly evaluate deductive/abductive/inductive reasoning, the responses first should have been turned into such problems and then LLMs re-queried to check for response matches. It's also been shown that LLMs don't necessarily perform multi-hop reasoning/answering, but infer a final response based on other context. To claim multi-hop reasoning, more evidence would be required.

The text of the paper, while overall well written and clear, also tends to overclaim, both in terms of reasoning conclusion without enough evidence, but in using terms like "exceptional multi-hop reasoning" (line 377).

Lastly, the comparative study against Chatbot Arena is too shallow and uses limited data. The conclusions attributed to it don't seem solid nor significant for the overall paper. Similar comments apply to the ranking comparisons in 4.5.

**Questions:**

Claims about reasoning, especially specific types of reasoning, require stronger evidence to support them. I'd expect to see a t least a couple case studies to support the claims presented in the evaluation. Not only some metrics and no detailed analysis.

If Table 2 lists GPT-4o as the best model for Taboo, why don't the metrics in Table 3 support that? The only discussion of the results for the Bluffing game also don't even mention how the described behaviour of other models relates to Claude 3.5 superior performance.

The calculation of disparity ratio in 3.2.2 is also not detailed enough. The authors claim the metric quantifies how effectively each step divides the answer space into two categories. But it only counts yes/no responses ratios?

In 4.1 the authors state the prompt optimization process uses 200 of the collected game sessions. Does this mean all data collection uses one set of prompts and the analysis uses the different optimized prompts?

Will the code used to calculate the different metrics be released openly? I feel this is a critical step for such benchmarks, as it would allow refining and standardization of definitions and calculations for re-use and reproducibility. Even if the direct mapping to specific reasoning categories describe is still not ideal.


As the comparison with Chatbot Arena and ranking comparisons don't seem to add much to support the main paper contributions, perhaps these could go into appendices to open space for more detailed analysis of the reasoning results.

---

> ### Author Response · Authors · 2024-11-19
>
> Thank you for your acknowledgment and valuable questions. Here are our responses to your questions and concerns:
>
> > W1: Firstly, the analysis of reasoning capabilities depends on a "replay" of a concluded game session round by round. At each round the replay prompt asks the models to output their "intermediary thought process". As LLMs are known for hallucination and fabricating rationalizations, this could heavily affect the proposed approach. There is no guarantee that whatever "evidence" being generated actually directly corresponds to the original inference rounds.  A more robust approach to collect the data might have been to ask for such output during the game.
>
> Asking LLMs to generate reasoning traces during gameplay could be an alternative approach. Which method to use is a trade-off between providing better user experience versus getting more reliable reasoning traces in real time. We didn’t use this method due to the following issues:
>
> 1. **Increased latency**: generating intermediary thoughts during gameplay will significantly slow down response times and disrupt gameplay experience.
>
> 2. **Potential disruption of game flow**: parsing exact answers from long, complex LLM responses is unreliable and may cause errors that break the game.
>
> Since GameArena relies on user participation to collect data, we prioritize user experience and chose our current method. In our future work, we plan to incorporate advanced reasoning techniques that can provide the reasoning traces during the game (similar to the o1 model). For now, we developed a simplified proof of concept.
>
> GameArena cannot fully eliminate hallucinations, as this remains an open challenge for all LLMs. However, our retrospective analysis keeps the context unchanged and is demonstrated reliable in many cases (see Appendix D for examples).
>
> > W2-1: The selected games also do not necessarily require the level of abstract reasoning being attributed to them in the paper. Akinator/Q20, for example, can be seen as a simple bisect search.
>
> We agree that a model that performs efficient bisect search is able to perform well on the akinator game. However, GameArena requires a good AI to reach the correct guess in as few steps as possible, which requires the model to ask questions strategically and guesses effectively.  Note that:
>
> - We have tested weaker models (e.g., GPT-3.5 Turbo, Open-Mixtral-8x22b, Claude-3.5-Haiku-Latest), which usually failed to identify objects within 20 rounds.
>
> - To evaluate this, we use one of our outcome metrics — the number of game rounds (Table 2). Winning in a smaller number of rounds implies a stronger reasoning capability.
>
> We will modify our paper and make these points clearer.
>
>
> > W2-2: Perhaps to properly evaluate deductive/abductive/inductive reasoning, the responses first should have been turned into such problems and then LLMs re-queried to check for response matches.
>
> We assume by “turned into such problems and then LLMs re-queried to check for response matches”, you are referring to the way users interact with AI during gameplay.
>
> GameArena wants to keep users incentivized and needs to support game-based natural conversation. As a result, we embed them within a broader context to reflect the model’s performance in a natural conversational setting for reasoning evaluation.
> We hope this addresses your concern, if not, we are willing to provide further explanation.
>
> > W2-3: It's also been shown that LLMs don't necessarily perform multi-hop reasoning/answering, but infer a final response based on other context. To claim multi-hop reasoning, more evidence would be required.
>
> > Q1: Claims about reasoning, especially specific types of reasoning, require stronger evidence to support them. I'd expect to see at least a couple case studies to support the claims presented in the evaluation. Not only some metrics and no detailed analysis.
>
> > W3: The text of the paper tends to overclaim, both in terms of reasoning conclusion without enough evidence, but in using terms like "exceptional multi-hop reasoning"
>
> We show that indeed in most gaming sessions, abductive/deductive/abductive and multi-hop reasoning are involved. For example, in Appendix D.3, the model uses information from all conversational rounds to gradually narrow down the possible range of words and eventually identifies the secret word. You can find more examples in Appendix D.1 and D.2. To strengthen this point, we will add case studies for a detailed analysis of reasoning results.

---

> ### Author Response · Authors · 2024-11-19
>
> > W4: the comparative study against Chatbot Arena is too shallow and uses limited data. The conclusions attributed to it don't seem solid nor significant for the overall paper. Similar comments apply to the ranking comparisons in 4.5.
>
> > Q6: As the comparison with Chatbot Arena and ranking comparisons don't seem to add much to support the main paper contributions, perhaps these could go into appendices to open space for more detailed analysis of the reasoning results.
>
> Our comparison goals are the following:
>
> 1. GameArena is a dynamic, incentivized benchmark focused on reasoning.
>
> 2. GameArena disentangles reasoning capabilities more effectively than dynamic benchmarks like Chatbot Arena by reducing confounding variables.
>
> 3. GameArena has a gamified interface that improves data collection efficiency (see section 4.2).
>
> The ranking comparison with Chatbot Arena provides evidence for the first two points (Table 4):
>
> 1. GameArena’s ranking aligns with other static reasoning benchmarks (LiveBench-Reasoning, GPQA).
>
> 2. GameArena’s ranking differs from Chatbot Arena as they are designed for different evaluation goals (Chatbot Arena for overall performance across a diverse set of real-world tasks).
>
> > Q2: If Table 2 lists GPT-4o as the best model for Taboo, why don't the metrics in Table 3 support that?
>
> GPT-4o has the highest win rate in Table 2 but is outperformed by Claude3.5-sonnect in Table 3 due to the following reasons:
>
> 1. We observed that GPT-4o tends to be cautious in responding to users’ questions (see examples in Appendix D.5 from GPT-4o’s responses). Refusals happen more often to GPT-4o than the other models, resulting in a higher win rate.
>
> 2. In terms of procedure metrics, GPT-4o is outperformed by Claude3.5-sonnect since claude-3.5-sonnet is more capable than gpt-4o in predicting the secret words.
>
> Therefore, evaluating models’ performance solely based on win rates might be problematic, which motivates us to design procedural metrics for a more comprehensive evaluation.
>
> >Q3: The calculation of disparity ratio in 3.2.2 is also not detailed enough. The authors claim the metric quantifies how effectively each step divides the answer space into two categories. But it only counts yes/no responses ratios?
>
> Given a list of possible objects and a set of questions to choose from, 20Q algorithms use yes/no disparity ratios to select the best question [1]. The idea is that an effective question should evenly divide the answer space into "yes" and "no" responses, halving the search space each time in a way similar to binary search. If most objects yield the same answer, the question provides little value to differentiate them. We build on this concept to develop a metric for evaluating the effectiveness of question generation.
>
> [1] Burgener, Robin. "Artificial neural network guessing method and game." U.S. Patent Application No. 11/102,105.
>
> > Q4: In 4.1 the authors state the prompt optimization process uses 200 of the collected game sessions. Does this mean all data collection uses one set of prompts and the analysis uses the different optimized prompts?
>
> To clarify, we initially use a hand-crafted system prompt to collect 200 game sessions. This data is then used to generate five optimized system prompts using DSPy. Both the data collection and analysis are based on these five optimized prompts.
>
> > Q5: Will the code used to calculate the different metrics be released openly?
>
> We have included the code for game implementation and data analysis in the supplementary material and available via [this link](https://anonymous.4open.science/r/game_arena-BB5E/data/taboo.json). We will also publicly open-source the code, game data, and analysis results.

---

> > ### Comment · Reviewer_kg44 · 2024-12-03
> >
> > Thank you very much for the responses to my review. While they clarify some of the issues raised, I still consider the additional information does not fully support the exact claims regarding the different reasoning categories.
> >
> > But I acknowledge authors addressed my concerns partially and I would like to increase my score to 6.

---

> ### Author Response · Authors · 2024-11-24
>
> Dear reviewer kg44,
>
> Thank you for taking the time to review our work. We would appreciate your feedback on the clarifications and additional analysis we provided:
>
> - W1: tradeoffs between using retrospective analysis versus collecting reasoning traces during gameplay.
>
> - W2, Q1, W3: How game sessions could involve sophisticated multi-hop, inductive, deductive, or abductive reasoning with case studies (Appendix D.1, D.2, D.3).
>
> - Q2: why win rates alone are insufficient as a metric for the Taboo game, as outcome metrics often conflict with procedural metrics.
>
> - W4, Q6: clarifications on the objectives of our comparisons and the key insights derived from ranking analyses.
>
> We are happy to address any further questions or concerns you may have.
>
> Best,
>
> Authors

---

> ### Author Response · Authors · 2024-11-27
>
> Dear reviewer kg44,
>
> As the discussion deadline approaches, we would like to ask you kindly review our responses and paper revisions. We are glad to provide any additional clarifications that you may need.
>
> Best,
>
> Authors

---

> > ### Author Response · Authors · 2024-11-30
> >
> > Dear Reviewer kg44,
> >
> > Thank you once again for your thoughtful and constructive feedback, as well as the time and effort you have dedicated to reviewing our submission! We believe addressing your comments have improved our work. As we approach the end of the rebuttal phase, would you please check our response and the revised version of our paper?
> >
> > We’re willing to discuss and address any additional feedback or concerns you may have.
> >
> > Best,
> >
> > Authors

---

> > > ### Author Response · Authors · 2024-12-03
> > >
> > > Dear Reviewer kg44,
> > >
> > > As today marks **the final day for follow-up questions** in the discussion period, we would like to take this final opportunity to address any remaining questions or concerns you may have regarding our response.
> > >
> > > Once again, we appreciate your time and effort in the review process.
> > >
> > > Best,
> > >
> > > Authors

---

### Author Response · Authors · 2024-11-19

Dear reviewers,

We thank all the reviewers for their feedback. We are encouraged by all reviewers' recognition that our benchmark is interesting (Reviewer kg44, JBob, 8gww) and comprehensive (Reviewer iRDM, JBob, 8gww). Specifically, we value the reviewers’ acknowledgment that GameArena:

1. carefully designs metrics for evaluating specific LLM reasoning abilities (Reviewer JBob, 8gww)

2. uses a gamified approach to enhance participant engagement (Reviewer kg44, JBob).

We have revised the paper and provided additional examples (in Appendix D) to illustrate:

1. cases in our games that involve sophisticated multi-hop, inductive, deductive, or abductive reasoning (reviewers kg44, JBob).

2. why win rates alone are insufficient as a metric for the Taboo game, as outcome metrics often conflict with procedural metrics (reviewers kg44, JBob).

3. case studies demonstrating the effectiveness of retrospective analysis (reviewer kg44). The results align sensibly with game session data.

In the rebuttal, we also included:

1. detailed game session statistics during the data collection time for both Chatbot Arena and GameArena for data efficiency comparison (reviewer JBob).
2. user-defined target word statistics (reviewer JBob).

3. annotator preference statistics (reviewer iRDM).

4. annotator demographic statistics (reviewer iRDM, JBob, 8gww).

Further details can be found in each individual response. We are happy to provide additional clarifications and address any other concerns the reviewers may have.

Best,

Authors

---

### Meta-Review · Area_Chair_GwFJ · 2024-12-19

**Metareview:**

The paper "GameArena: Evaluating LLM Reasoning through Live Computer Games" introduces a novel dynamic benchmark, GameArena, designed to evaluate the reasoning capabilities of large language models (LLMs) through interactive gameplay with humans. The benchmark comprises three distinct games, each tailored to assess specific reasoning skills such as deductive, inductive, and abductive reasoning. The authors collect over 2000 game sessions and provide a detailed assessment of five state-of-the-art LLMs, demonstrating GameArena's ability to provide fine-grained reasoning evaluations. A user study with 100 participants indicates that GameArena enhances user engagement compared to existing benchmarks like Chatbot Arena, while also enabling the collection of step-by-step reasoning data in real-world interactions for the first time.

The primary contributions of the paper are:
1. **GameArena Benchmark**: A novel, dynamic, and engaging benchmark for evaluating LLM reasoning capabilities through interactive games, offering a fresh approach compared to static datasets prone to contamination and saturation.
2. **Detailed Reasoning Assessment**: Comprehensive evaluation of five state-of-the-art LLMs, providing insights into their deductive, inductive, and abductive reasoning abilities through both outcome and procedural metrics.
3. **Enhanced User Engagement**: Empirical evidence from a user study demonstrating that GameArena improves participant engagement compared to existing benchmarks, which could facilitate more effective data collection in the future.

### Weaknesses

1. **Dataset and Analysis Limitations**: Some reviewers raised concerns about the robustness of the retrospective analysis method used to uncover reasoning processes, suggesting that collecting reasoning traces during gameplay might be more reliable. Additionally, the limited sample size (n=5 models) in comparative analyses with other benchmarks was noted as potentially insufficient for statistical significance.

2. **Clarity and Overclaiming**: The paper's descriptions of game rules and reasoning processes were deemed unclear or overly ambitious by some reviewers. For instance, the claimed levels of abstract reasoning in the games were questioned, and the language used in the manuscript was perceived as overclaiming in some instances.

3. **Comparative Study Depth**: The comparison with Chatbot Arena was criticized for being superficial, with limited data and insufficient depth to draw robust conclusions. Similarly, the ranking comparisons lacked detailed statistical validation, which could undermine the reliability of the findings.

4. **Presentation Issues**: Some minor presentation issues were noted, including ambiguous phrases, inconsistent table layouts, and missing statistical details, which could impact the manuscript's readability and clarity.

5. **Lack of Broader Context and Limitations Discussion**: The paper did not adequately discuss potential limitations, such as biases in user-generated topics or the diversity of participant demographics, which could affect the benchmark's generalizability. Additionally, the authors' claim of being the first to apply gamified approaches to LLM benchmarking was questioned, as similar methods have been explored in other fields.

**Additional Comments On Reviewer Discussion:**

1. **Retrospective Analysis vs. Real-Time Collection (kg44):**
   - **Concern**: The reliance on retrospective analysis for reasoning traces was criticized due to potential LLM hallucinations and the lack of real-time data collection.
   - **Response**: The authors explained the trade-offs, highlighting that real-time collection could disrupt gameplay and user experience. They provided examples and case studies to demonstrate the reliability of their retrospective analysis and committed to exploring real-time reasoning techniques in future work.
   - **Impact**: While the concern was partially addressed, the reviewer remained unconvinced about the robustness of the approach, leading to a maintained score of 6.

2. **Game Complexity and Reasoning Claims (kg44):**
   - **Concern**: The games were deemed insufficiently complex for the claimed reasoning skills, and the paper was criticized for overclaiming reasoning capabilities.
   - **Response**: The authors provided additional examples and case studies in the appendix to illustrate the reasoning processes involved and clarified the metrics used to evaluate reasoning skills.
   - **Impact**: The additional evidence and clarifications were acknowledged, though the reviewer noted that the claims were still not fully supported, maintaining a score of 6.

3. **Comparative Study with Chatbot Arena (kg44, JBob):**
   - **Concern**: The comparison with Chatbot Arena was seen as shallow and lacking in depth, with concerns about the fairness and significance of the data efficiency comparison.
   - **Response**: The authors clarified the goals of the comparison, provided additional game session statistics, and emphasized the different evaluation objectives of GameArena and Chatbot Arena.
   - **Impact**: The responses clarified the comparison's purpose, but the reviewers still found the analysis lacking depth, leading to maintained scores around 6.

4. **Metric Design and Interpretation (JBob, kg44):**
   - **Concern**: The design and interpretation of metrics, particularly the disparity ratio and win rates, were questioned for their effectiveness in evaluating reasoning.
   - **Response**: The authors provided detailed explanations of the metrics' design and rationale, supported by references and examples, and emphasized the need for both outcome and procedural metrics for comprehensive evaluation.
   - **Impact**: The clarifications were well-received, with reviewers acknowledging the thoughtful metric design, though some concerns about interpretation remained.

5. **User Demographics and Game Statistics (iRDM, 8gww):**
   - **Concern**: The lack of detailed participant demographics and game statistics raised concerns about potential biases and the benchmark's generalizability.
   - **Response**: The authors provided comprehensive demographic and game statistics, addressing potential biases and diversity issues.
   - **Impact**: The detailed statistics addressed the reviewers' concerns, with one reviewer (iRDM) expressing full satisfaction and maintaining a high score of 8.

6. **Future Directions and Broader Context (8gww):**
   - **Concern**: The paper did not sufficiently discuss potential limitations or provide insights for future research, limiting its broader context.
   - **Response**: The authors acknowledged limitations related to topic distribution and participant diversity, outlined future research directions, and provided insights from their evaluation results.
   - **Impact**: The inclusion of limitations and future directions was appreciated, though one reviewer (8gww) suggested further statistical validation for comparative analyses, maintaining a score of 6 but leaning towards acceptance.

**Final Decision Weighting:**
The paper introduces a novel and engaging benchmark for evaluating LLM reasoning, with a well-designed methodology and comprehensive evaluation of state-of-the-art models. The authors' responsiveness to feedback, providing additional examples and clarifications, demonstrates a commitment to improving the manuscript. However, concerns about the robustness of the retrospective analysis, the depth of comparative studies, and the clarity of reasoning claims persist.

---

### Decision · Program_Chairs · 2025-01-22

Accept (Poster)